# Aerodynamic characterization of a soft kite by in situ flow measurement

Johannes Oehler[1] and Roland Schmehl[1]

[1]Faculty of Aerospace Engineering, Delft University of Technology, 2629 HS Delft, Netherlands

*Correspondence to:* Roland Schmehl r.schmehl@tudelft.nl

**Abstract.** Wind tunnel testing of large deformable soft kites for wind energy conversion is expensive and in many cases practically not feasible. Computational simulation of the coupled fluid-structure interaction problem is scientifically challenging and of limited practical use for aerodynamic characterization. In this paper we present a novel experimental method for aerodynamic characterization of flexible membrane kites by in situ measurement of the relative flow, while performing complex flight maneuvers. We find that the measured aerodynamic coefficients agree well with the values that are currently used for flight simulation of soft kites. For flight operation in crosswind maneuvers where the traction force is kept constant, the angle of attack is inversely related to the relative flow velocity. For entire pumping cycles, the measurements show considerable variations of the aerodynamic coefficients, while the angle of attack of the kite varies in fact only in a narrow range. This finding questions the commonly used representation of aerodynamic coefficients as sole functions of the angle of attack and stresses the importance of aeroelastic deformation for this type of wing. Considering the effect of the power setting (identical to the trim) solely as a rigid-body pitch rotation does not adequately describe the aero-structural behavior of the kite. We show that the aerodynamic coefficients vary as functions of the power setting (trim) of the kite, the steering commands and flight direction.

## 1 Introduction

Airborne wind energy is the conversion of wind energy into electrical or mechanical power by means of flying devices. Some of the pursued concepts use tethered airplanes or gliders, while others use flexible membrane wings that are derived from surf kites or parafoils (Diehl et al., 2017). The present paper is focusing on an airborne wind energy system (AWES) with an inflatable membrane wing that is controlled by a suspended cable robot (van der Vlugt et al., 2013, 2019). Compared to rigid-wing aircraft, the aerodynamics of tethered-membrane wings are not so well understood and kite development still relies heavily on subjective personal experience and trial and error processes (Breukels, 2011; Dunker, 2013). One reason for this is the aeroelastic two-way coupling of wing deformation and air flow which can cause complex multi-scale phenomena. Another reason is a lack of accurate quantitative measurement data to support the design process. Soft kites such as leading edge inflatable (LEI) tube kites are highly flexible and have no rigid structure to mount sensors for precise quantification of the relative flow in the vicinity of the wing. This is why many experiments rely on ground-based force measurements and position

tracking of the kite. In these experiments the environmental wind velocity introduced substantial uncertainties (Python, 2017; Hummel et al., 2018).

With dimensions in the order of several meters, large surf kites or even larger kites for power generation exceed the size capacity of most wind tunnels. Downscaling a physical model, as it is customary for rigid-wing aircraft, would require a synchronous scaling of the aerodynamic and structural problems, which for a fabric membrane structure with seams, wrinkles, multiple functional layers and integrated reinforcements is practically very difficult, if not unfeasible. For example, scaled models of large gliding parachutes have been analyzed in the wind tunnel at NASA Ames Research Center (Geiger and Wailes, 1990), while a 25% model of the FASTWing parachute was tested in the European DNW-LLF wind tunnel (Willemsen et al., 2005). A first full-scale experiment to determine the shape of a kite in controlled flow conditions was performed by de Wachter (2008). Using photogrammetry as well as laser light scanning the three-dimensional surface geometry of a small ram-air surf kite was measured in two larger wind tunnels. This geometry was used as boundary condition for computational fluid dynamic (CFD) analysis of the exterior flow. The results show a substantial deformation of the membrane wing by the aerodynamic loading. Due to the difficulty of scaling, these results can not be transferred to larger kites for wind energy conversion that fly at higher speeds.

In general, the numerical simulation of strongly coupled fluid-structure interaction (FSI) problems is computationally expensive. If the flow is fully attached, standard panel methods with viscous boundary layer models can be used for efficient calculation of the aerodynamic load distribution. While this approach works, for example, for ram-air wings at lower angle of attack, it is not feasible for LEI tube kites because of the inevitable flow separation region behind the leading edge tube. Breukels (2011) and Bosch et al. (2014) develop multibody and finite element models of LEI tube kites and use an empirical correlation framework to describe the aerodynamic load distribution on the membrane wing as function of shape parameters. Bungart (2009) performs CFD analysis using the deformed shape of the kite measured by de Wachter (2008), however, these results can not be extrapolated to different kites. We conclude that without accounting for the aeroelasticity of the membrane wing an accurate aerodynamic characterization does not seem to be feasible. We further conclude that presently experiments seem to be the most viable option to determine the global aerodynamic characteristics of a kite.

In Table 1 we list experiments described in literature to determine the lift-to-drag ratio of kites. The relative flow velocity at the wing is denoted as $v_a$ and the power setting $u_p$ describes the symmetric actuation of the rear suspension lines of the kite. A high value of $u_p$ means that the wing is powered, while a low value of $u_p$ means that the wing is depowered (see Eq. 6). The variety of methods, test conditions and kites as well as generated results makes it difficult to derive a clear trend. Hobbs (1990) conducted a performance study of different single-line kite designs used for wind anemometry. A first quantitative aerodynamic assessment method for power kites was presented by Stevenson (2003), Stevenson et al. (2005) and Stevenson and Alexander (2006). The test procedure involves flying kites on a circular trajectory indoors as well as outdoor towing tests.

A similar manual test procedure for determining the lift-to-drag ratio of a surf kite was proposed by van der Vlugt (2009). The kite is flown in horizontal crosswind sweeps just above the ground, measuring the achievable maximum crosswind flight speed $v_{k,\tau}$ of the kite at a downwind position together with the wind speed $v_w$. Assuming that the measured wind speed is

**Table 1.** Experimental methods for determining the lift-to-drag ratio of soft kites. Size refers here to total wing surface area.

| method | kite type | size [m$^2$] | limitations | wing loading [N/m$^2$] | $v_a$ [m/s] | relative power setting $u_p$ [-] | $L/D$ [-] | reference |
|---|---|---|---|---|---|---|---|---|
| rotating arm | C-Quad | 3.2 | kite size, forces | 100 | 11 | low | 4.9 | Stevenson et al. (2005) |
| towing test | C-Quad | 3.2 | unknown wind | – | – | low | 4.6–5.6 | Stevenson et al. (2006) |
| wind tunnel | ram air | 6 | kite size | 25 | 8 | low–high | 6 | de Wachter (2008) |
| wind tunnel | ram air | 6 | kite size | 60 | 12 | low–high | 6.7–5.7 | |
| wind tunnel | ram air | 6 | kite size | 120 | 16 | low–high | 8–5.5 | |
| crosswind | ram air | 6 | kite size, forces | 300 | 24 | high | 6.1 | van der Vlugt (2009) |
| towing test | ram air | 3 | kite size, forces | 30 | 8 | – | 6 | Dadd et al. (2010) |
| towing test | LEI | 15.3 | force/speed limited | 40 | 14 | – | 4.5–5.5 | Costa (2011) |
| crosswind | LEI | 14 | wind data unknown | 140 | – | high | 6 | Ruppert (2012) |
| towing test | LEI | 14 | force/speed limited | 40 | 11.3 | low–high | 4–10 | Hummel (2017) |
| crosswind | LEI | 5 | kite size | 300 | 20 | high | 4.6 | Behrel et al. (2018) |
| crosswind | LEI | 14, 25 | wind data unknown | 215, 123 | – | high | 4, 3.6 | van der Vlugt et al. (2019) |

identical with the wind speed at the kite, the lift-to-drag ratio can be calculated from (Schmehl et al., 2013)

$$v_{k,\tau} = \frac{L}{D} v_w. \tag{1}$$

The method can be generalized to characterize the aerodynamics of kites flying complex maneuvers by either measuring or estimating the unperturbed relative flow velocity $v_a$ in the vicinity of the wing. Figure 1 shows a self-aligning Pitot tube setup mounted in the bridle line system between kite and its control unit. The placement of the Pitot tube in the bridle line system

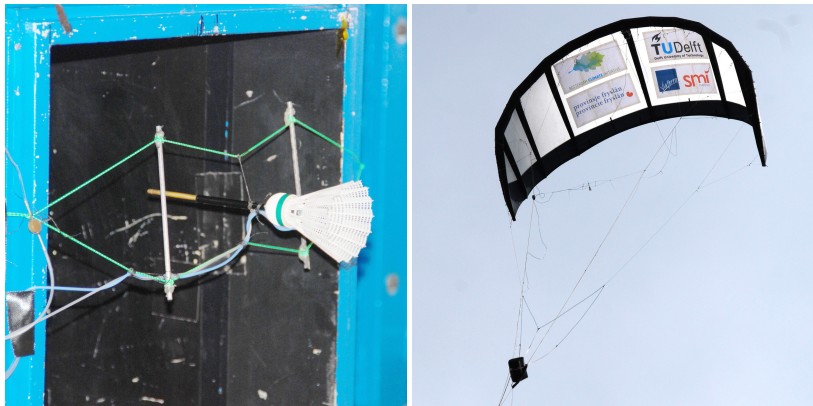

**Figure 1.** Pitot tube during calibration in the wind tunnel (left) and suspended in the bridle line system of a remote-controlled 25 m$^2$ LEI V2 kite during a flight test (right).

was chosen to avoid a perturbation of the relative flow by the wing and the control unit. However, Ruppert (2012) concluded that the quality of the measurement data of this setup was insufficient and thus estimated the wind speed at the kite from other available data. In lack of reliable velocity measurements, van der Vlugt et al. (2019) describe an approach to estimate the lift-to-drag ratio of the airborne system components from measured force and position data. Borobia et al. (2018) have mounted a Pitot tube directly on the center strut of a small surf kite to measure the relative flow speed. Together with the data of other onboard sensors, this has been used to feed an extended Kalman filter to get an optimal estimate of the aerodynamic force and torque generated by the kite as well as the relative flow velocity vector and other kite state variables.

Dadd et al. (2010) and Costa (2011) used towing test setups to generate a variable relative flow at the wing. Operating at days with calm wind allows for measurements at well-defined relative flow conditions. Hummel et al. (2018) developed a similar trailer-mounted towing test setup to measure the lift-to-drag ratio and aerodynamic coefficients of surf kites. The test procedure includes active depowering, which, in general aerospace engineering terminology is denoted as a change in trim of the wing and measuring line angles at the test rig. For future experiments, Hummel recommends the use of an airborne flow sensor to avoid the uncertainties caused by the wind environment and by the sagging of the tether. Behrel et al. (2018) describe a setup to measure the aerodynamic performance of kites for ship traction applications. Using a three-dimensional load cell to record the traction force vector and a wind profiler to determine the wind velocity at the kite, the technique is applied to determine the lift-to-drag ratio of kites during crosswind maneuvers.

The companies Kitepower B.V., a startup of Delft University of Technology, Kite Power Systems (KPS) and Skysails Power (Weston, 2018) are currently developing and testing AWES with soft kites that are operated on a single tether and controlled by a suspended cable robot. These prototypes have reached considerable sizes (see for example Fig. 2) and for this reason the use of measurement data acquired during flight operation is the only viable option for characterizing the aerodynamics of the complete airborne system. None of the other experimental setups presented in Table 1 allows to execute dynamic flight

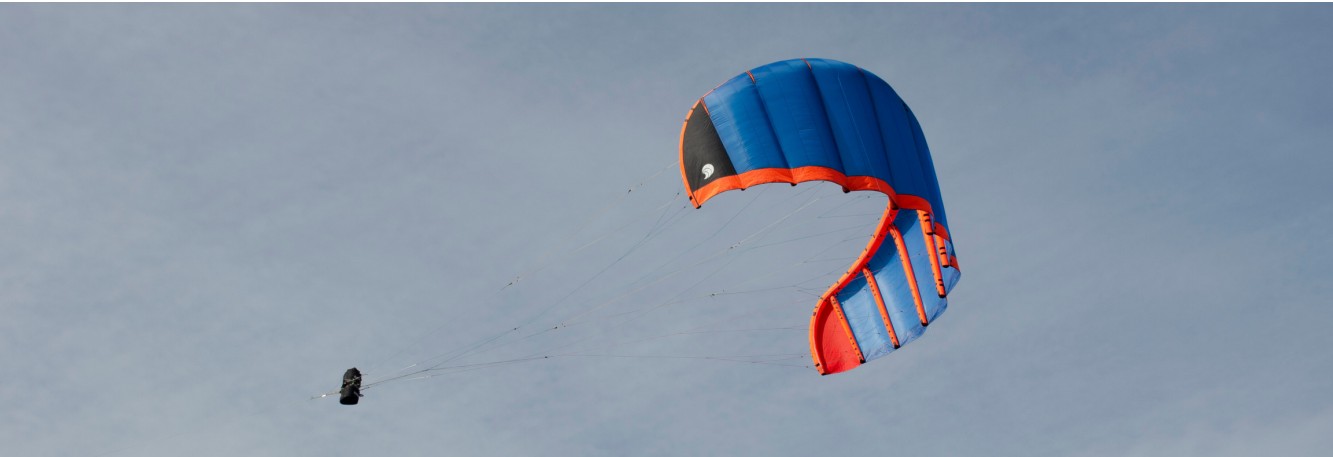

**Figure 2.** LEI V5.40 kite with 40 m$^2$ wing surface area controlled by a suspended cable robot. This prototype temporarily reached a tether force of 15 kN and a mechanical power of 100 kW during a test flight in May 2018 (Kitepower B.V., 2018).

maneuvers and handle kites with a wingspan of 10 m or larger, at flight speeds above 20 m/s while withstanding tensile forces of several kN or more. It is the objective of the present study to develop an experimental method for aerodynamic characterization of large deformable membrane kites that are used for energy conversion. At the core of this method is a novel setup for the accurate measurement of the relative flow conditions at the kite during energy-generation in pumping cycles. Since the setup

5    is additional equipment for tests of a commercial prototype the mounting of the setup has to consume as little time as possible.

The paper is organized as follows. In Sect. 2 we describe the airborne components of the kite power system, the measurement setup and the data acquisition procedure. In Sect. 3 we describe how the power setting is related to the angle of attack of the wing and how the aerodynamic properties are derived from the measured data. In Sect. 4 the results are presented and discussed.

## 2    System description and data acquisition

10    The experimental study is based on the AWES prototype developed and operated by the company Kitepower as a test platform within the EU Horizon 2020 "Fast Track to Innovation" project REACH (European Commission, 2015). The prototype can be classified as a ground-generation AWES, operating a remote-controlled soft kite on a single tether. This general setup is illustrated schematically in Fig. 3 (right). The main system components are the ground station for converting the linear

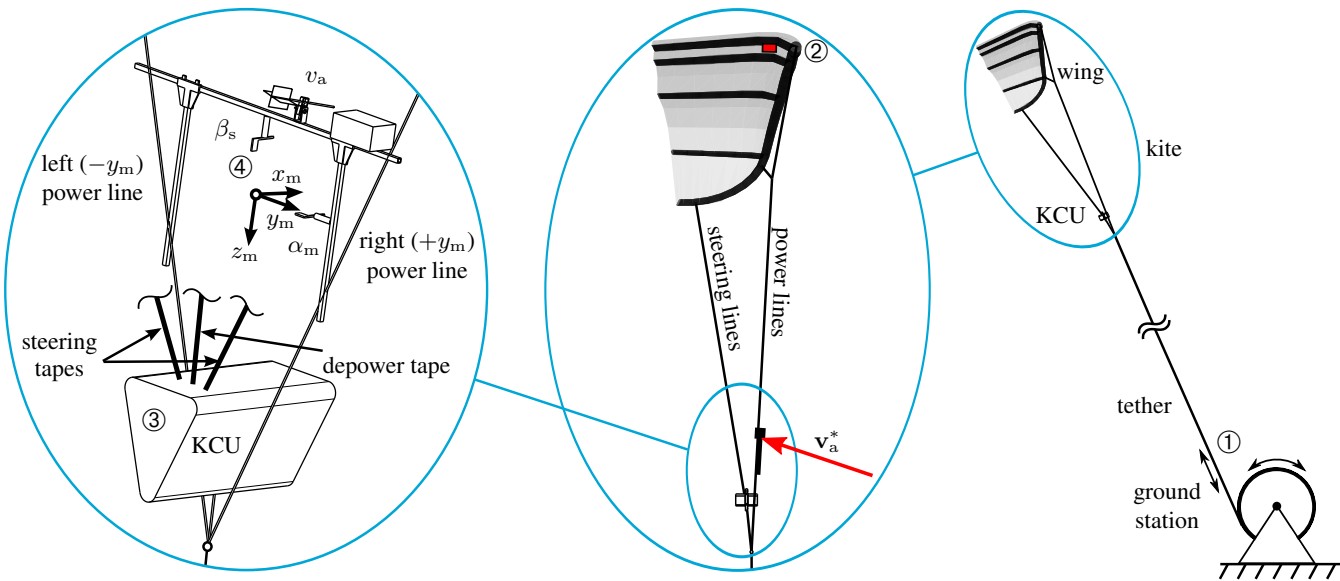

**Figure 3.** Basic components of the kite power system (right), wing with suspended control unit, together denoted as kite (center), and measurement frame attached to the power lines (left). Sensor positions: tether force $F_t$ and tether reel-out speed $v_t$ are recorded at the ground station ①, GPS and IMU modules are mounted on the center strut of the kite ②, the kite control unit ③ actuates the wing for steering and changing its power state, measuring also the instantaneous lengths of steering and depower tapes, the relative flow sensors for inflow angles $\alpha_m$, $\beta_s$ and apparent wind speed $v_a$ are mounted on a rigid frame ④ that is attached to the two power lines connecting to the inflatable leading edge tube of the wing. The depicted velocity $\mathbf{v}_a^*$ is the component of the apparent wind velocity $\mathbf{v}_a$ projected into the drawing plane.

traction motion of the kite into electricity, the main tether and the C-shaped, bridled wing with the suspended kite control unit (KCU). In the following, we will denote the assembly of wing, bridle line system and KCU as "kite". To generate power the kite is operated in cyclic flight patterns with alternating traction and retraction phases. During the traction phase the kite performs crosswind maneuvers, such as figure-of-eight or circular flight patterns, while the tether is reeled off a drum that is connected to a generator. In this phase the AWES generates electricity. For the subsequent retraction phase the crosswind maneuvers are stopped and the generator is operated as a motor to reel in the tether. This phase consumes some of the generated electricity. To maximize the net gain of energy per cycle the wing is depowered during retraction. Both steering lines are released symmetrically such that the entire wing pitches down, to a lower angle of attack, which significantly reduces the aerodynamic lift force.

Just below the KCU the main tether splits into two power lines of constant length that run along the sides of the KCU and support the inflatable leading edge tube and partially also the strut tubes of the wing. This is depicted schematically in Fig. 3 (center and left) and in more detail in Fig. 5 (left) without the measurement setup. A short line segment connects the KCU to the end point of the main tether, while steering and depower tapes connect the KCU to the steering lines and eventually, via a fan of bridle lines, to the wing tips and trailing edge. Details on this specific layout will be described in the following section. The KCU can actuate the two steering lines either symmetrically, to power and depower the kite, or asymmetrically, to steer the kite. The actuation of the wing as part of the kite is illustrated in Fig. 4. The photographic footage from 23 August 2012

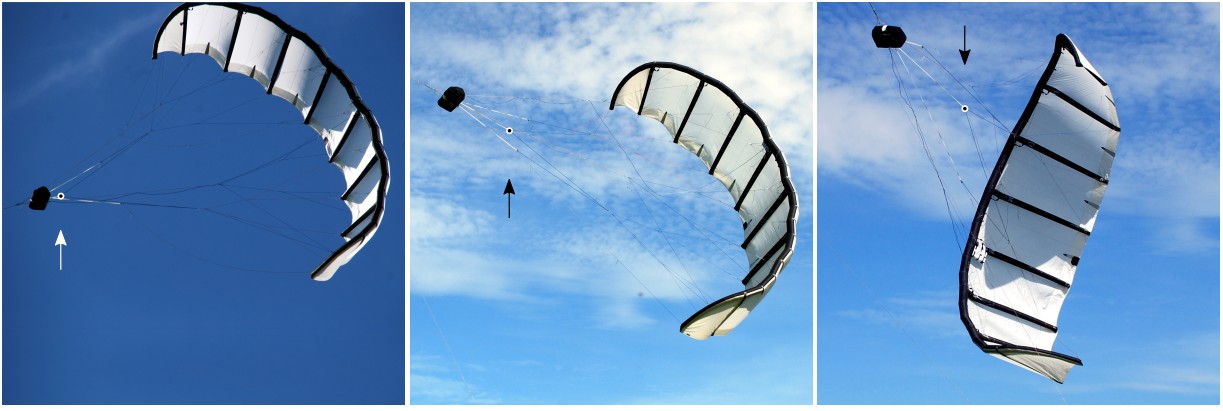

**Figure 4.** Almost fully powered LEI V3 kite (left), depowered kite (center) and deformation of the wing by extreme steering input in depowered state (right). Dots indicate the end of the depower tape.

is documenting tests of a mast-based launch setup. While the left photo is taken during crosswind maneuvers during power generation, the two right photos are taken during a flight maneuver close to the launch mast.

The sensors on the ground station ①, the kite ② and the KCU ③ provide data that is required for the autopilot of the kite power system (see Fig. 3). The experiments described in this paper have been performed with a LEI V3 kite with a wing surface area of 25 m$^2$, a battery-powered KCU for 2–3 hours of continuous operation and a ground station with 20 kW nominal traction

power. These components have been developed by the kite power research group of Delft University of Technology and reflect the technology status in 2012 (van der Vlugt et al., 2013; Schmehl, 2014; Schmehl et al., 2014; van der Vlugt et al., 2019).

Because the membrane wing is continuously deforming during operation it is not as straightforward as for a rigid-wing aircraft to define the orientation of the kite relative to the flow. One option is to use the inflated center strut as a reference component to mount the flow measurement equipment (van Reijen, 2018; Borobia et al., 2018). Mounting the equipment directly on the suspended KCU is not considered to be an option because this relatively heavy component is deflected substantially when flying sharp turns (Fechner and Schmehl, 2018) and can also exhibit transverse vibrations. Another option is to mount the measurement equipment on the two power lines. These lines transfer the major part of the aerodynamic force from the wing to the tether and for this reason are generally well-tensioned and span a plane that characterizes the orientation of the kite (wing and suspended KCU). Considering the deformation of the membrane wing by asymmetric and symmetric actuation as well as aeroelasticity, we consider this plane to be the most suitable reference geometry.

Figure 3 (left) illustrates how the three relative flow sensors ④ are mounted on a rigid frame that is attached to the two power lines about 8.5 m below the wing. In Appendix A we use a simple lifting-line model of the wing to show that the assumption of free stream conditions at this distance from the kite is justified. The Pitot tube can rotate freely around its pitch and yaw axis to align with the relative flow, measuring the barometric pressure, the differential pressure and the temperature from which the apparent wind velocity $v_\mathrm{a}$ can be calculated. The two flow vanes are used to determine the inflow angles $\alpha_\mathrm{m}$ and $\beta_\mathrm{s}$ which are measured from the normal vector of the plane spanned by the two power lines. The two angles are measured by total magnetic encoders with a resolution of $0.35°$. The data is recorded at a frequency of 20 Hz, converted to a digital signal by a Pixhawk® microcomputer, transmitted to the KCU via antenna and from there to the ground station to be logged simultaneously with all other acquired sensor data. It is important to note that the relative flow sensors are add-on measurement equipment for the present study and are not essential for the operation of the kite power system. More information on the sensors and the measurement setup can be found in Oehler (2017).

The new setup addresses two shortcomings of the earlier attempts to determine the relative flow conditions at a kite, illustrated in Fig. 1. Firstly, a self-aligning Pitot tube alone can measure only the magnitude of the relative flow velocity but not its direction. The orientation of the wing relative to the flow is however important information for the aerodynamic characterization. Secondly, the tensile suspension of the Pitot tube in the bridle system of the kite was not sufficiently robust against perturbations which negatively affected the quality of the measurement results. Jann and Greiner-Perth (2017) describe a similar setup for gliding parachutes, mounted in the bridle lines between payload and wing, to measure the angle of attack and relative flow speed. By choosing a setup that is flying with the kite we are able to acquire the relative flow conditions in situ, during operation of the full-scale system, and are not constrained by the traction force limit of a particular ground testing rig. This allows us to characterize also the aerodynamics of power kites that produce much more lift force than usual surf kites.

## 3 Data processing

The raw data from the rotary encoders and the pressure sensor can have missing data points and can also fluctuate due to variations of the supply voltage. To address these issues we apply a moving-average filter, using the Matlab® function *smooth* with a span of 7 measurement points (0.3 s). This operation returns a smooth signal while still being able to capture systematic oscillations that occur at frequencies of up to 1.2 Hz (Oehler, 2017). In the following, we describe how the relative flow data is used together with the data of the other sensors to determine the aerodynamic properties of the kite.

### 3.1 Geometry and reference frames

The geometry of the wing and the layout of the tensile support system, comprising bridle lines, steering and depower lines, as well as steering and depower tapes are illustrated in Fig. 5. The two pulleys are attached to the two branches of the rear bridle line systems and allow the steering lines to slip freely to adjust the line geometry to the actuation state. The instantaneous length of the depower tape is denoted as $l_d$. Both renderings show a depowered kite, as illustrated by the photo in Fig. 4 (center), using the design shape (CAD geometry) of the wing and thus not accounting for additional deformation.

As shown in Fig. 5 (right), we define two different reference frames to describe the orientation of the tether and the kite. The tether reference frame $(x_t, y_t, z_t)$ is attached to the kite end of the tether with its origin at the bridle point **B** where the tether splits into the two power lines. The $z_t$-axis is tangential to the tether, while the $x_t$-axis is located in the plane spanned by the $z_t$-axis and the normal vector of the plane spanned by the two tensioned power lines. This definition is identical to the "kite reference frame" used by Fechner et al. (2015) as a basis for a point mass model. The measurement reference frame $(x_m, y_m, z_m)$ is attached to the rigid frame on which the relative flow sensors are mounted. As depicted in Fig. 3 (left), the $z_m$-axis is aligned with the two upright members of the frame, while the $y_m$-axis is aligned with the transverse member. Because the measurement frame is attached to the two tensioned power lines the $x_m$-axis defines the heading of the kite. The rotation of the $x_t$-axis into the $x_m$-axis is described by the angle $\lambda_0$, which is not constant and can not be controlled actively. The angle depends on the aerodynamic load distribution acting on the wing, the kite design and the bridle layout. The inflow angles $\beta_s$ and $\alpha_m$ are determined in the measurement reference frame. Because the $z_m$-axis can be regarded as the yaw axis of the kite, the inflow angle $\beta_s$ is equivalent to the side slip angle. Similarly, the $y_m$-axis can be regarded as the pitch axis of the kite and the inflow angle $\alpha_m$ is a measure for its pitch orientation with respect to the relative flow.

To transform $\alpha_m$ into a meaningful angle of attack of the wing we define a reference chord $c_{ref}$ which describes the pitch orientation of the wing within the kite system as a function of the symmetric actuation of the steering lines. This two-dimensional, simplified geometric depower model is illustrated in Fig. 5 (right). For the fully powered kite, the reference chord is defined to be perpendicular to the plane spanned by the power lines. Depowering the kite is modeled as a pitching of the reference chord around the front suspension point, while the real wing additionally deforms by spanwise twisting and bending. The specific bridle layout of the LEI V3 kite shifts the front suspension point about 0.5 m backwards from the leading edge. The rotation is described by the depower angle $\alpha_d$ and by definition the fully powered state is given by $\alpha_d = 0$. A reference chord that is perpendicular to the power line plane is a reasonable approximation of the fully powered wing which is designed for optimal

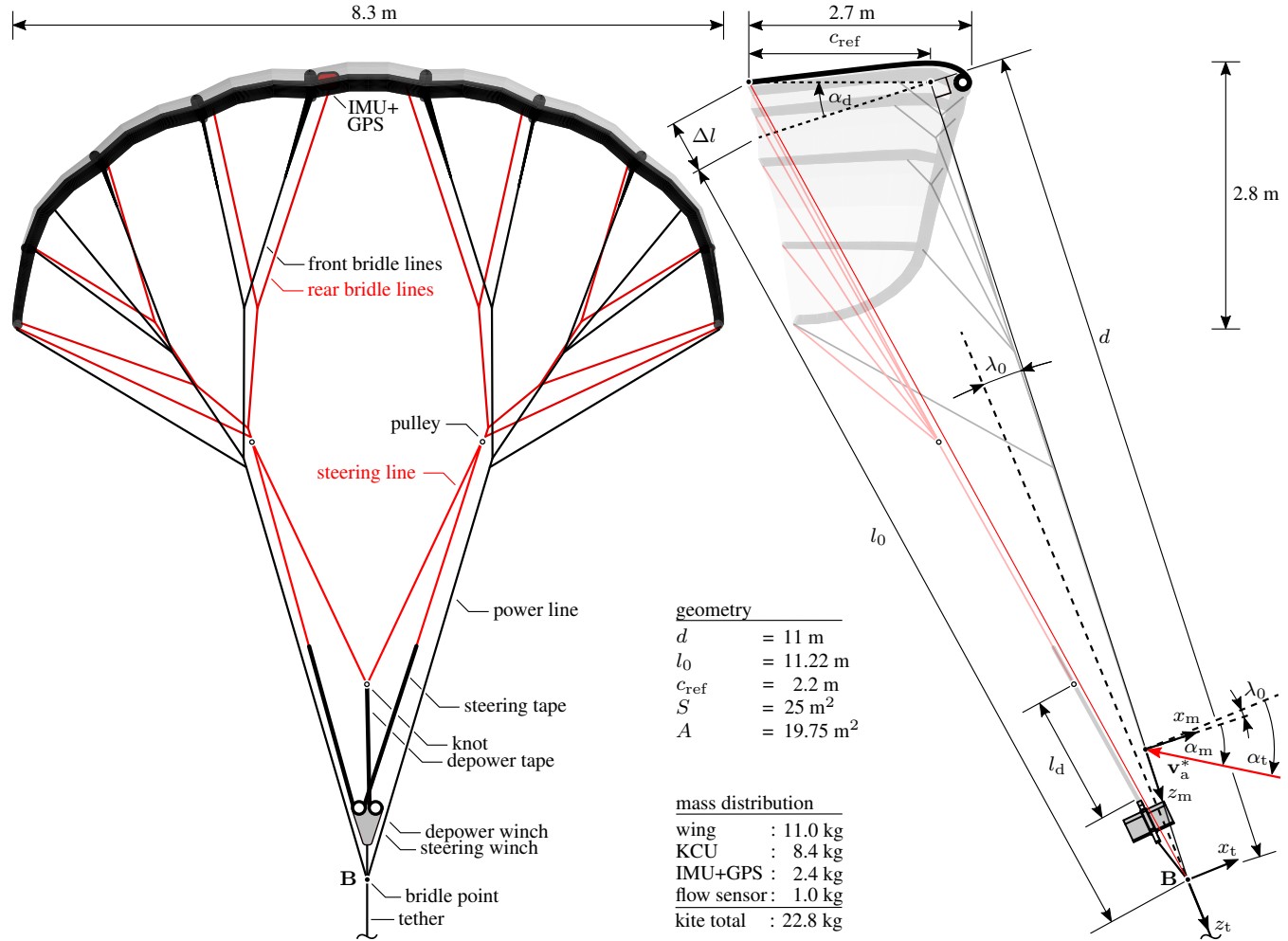

**Figure 5.** Front view (left) and side view (right) of the LEI V3 kite with reference frames, geometric parameters, mass distribution and definition of the reference chord $c_{\text{ref}}$. The total wing surface area is denoted as $S$, while the projected value is denoted as $A$. The mass of the bridle lines is part of the wing mass. The side view distinguishes between the physical (real) kite and bridle line system, displayed in the background, and the overlaid simplified geometric depower model. The explicit dimensions describe the unloaded design shape of the wing.

transfer of the aerodynamic load from the membrane wing to the bridle line system. These structural requirements are generally met best if the front bridle lines, which transmit most of the forces, connect perpendicularly to the wing. It is in principle straightforward to account for a constant offset angle $\alpha_0$ (Fechner et al., 2015), however, for the investigated kite design this offset angle is rather small. For this reason we set $\alpha_0 = 0$.

5    The geometrical dimensions are extracted from the CAD geometry of the kite. The distance of the front suspension point from the bridle point is $d = 11.0$ m. For the fully powered kite, the distance of to the rear suspension point from the bridle point is $l_0 = 11.22$ m. The length of the reference chord can be determined as $c_{\text{ref}} = 2.2$ m. The kite is depowered by extending the

rear suspension of the wing by $\Delta l$. In the following section, we relate this length extension to the deployed length $l_{\mathrm{d}}$ of the depower tape and the relative power setting $u_{\mathrm{p}}$. The angle of attack of the relative flow with respect to the reference chord is calculated from the measured inflow angle and the depower angle as

$$\alpha = \alpha_{\mathrm{m}} - \alpha_{\mathrm{d}}, \tag{2}$$

5    while the angle of attack of the relative flow with respect to the tether reference frame is calculated as

$$\alpha_{\mathrm{t}} = \alpha_{\mathrm{m}} + \lambda_0. \tag{3}$$

Figure 6 illustrates how the azimuth angle $\phi$, the elevation angle $\beta$ and the radial distance $r$ are used to specify the position of the bridle point $\mathbf{B}$ relative to the ground attachment point $\mathbf{O}$. The heading angle $\psi$ specifies the orientation of the kite in

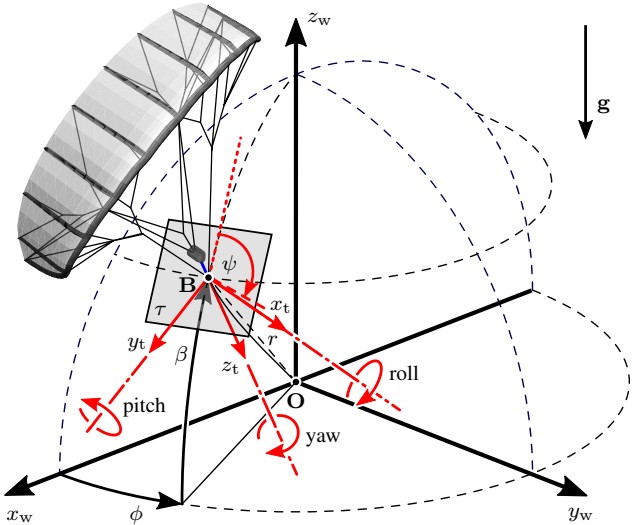

**Figure 6.** Ground reference frame $(x_{\mathrm{w}}, y_{\mathrm{w}}, z_{\mathrm{w}})$, tether reference frame $(x_{\mathrm{t}}, y_{\mathrm{t}}, z_{\mathrm{t}})$, heading angle $\psi$ and spherical coordinates $(\beta, \phi, r)$. Only in case of a straight tether, the $z_{\mathrm{t}}$-axis is pointing in radial direction to the ground attachment point $\mathbf{O}$.

the local tangential plane $\tau$. The angle is measured between the local upward direction (dotted line) and the projection of the 10    $x_{\mathrm{t}}$-axis onto the tangential plane. Similarly, the course angle $\chi$ (not displayed) specifies the direction of the tangential kite velocity $\mathbf{v}_{\mathrm{k}, \tau}$ in the local tangential plane. Combining Eqs. (2) and (3) to eliminate the measured inflow angle $\alpha_{\mathrm{m}}$ we can differentiate three distinct contributions to the angle of attack

$$\alpha = \alpha_{\mathrm{t}} - \lambda_0 - \alpha_{\mathrm{d}}. \tag{4}$$

The contribution of the tether angle of attack $\alpha_{\mathrm{t}}$ is due to the flight motion of the kite, represented by the bridle point $\mathbf{B}$, 15    through the wind environment. The contribution of the line angle $\lambda_0$ is due to the pitch of the entire kite, represented by the plane spanned by the power lines, with respect to the tether. The contribution of the depower angle $\alpha_{\mathrm{d}}$ is due to the pitch of the wing with respect to the plane spanned by the power lines..

### 3.2 Kinematics of depowering

Instead of assuming a linear correlation between the relative power setting $u_p$ and the depower angle $\alpha_d$, as proposed by Fechner et al. (2015), we use the geometric depower model illustrated in Fig. 5 (right) to calculate an analytic equation for the depower angle $\alpha_d$ by applying the law of cosines

$$\cos(90° + \alpha_d) = \frac{d^2 + c_{\mathrm{ref}}^2 - (l_0 + \Delta l)^2}{2 d c_{\mathrm{ref}}}. \tag{5}$$

Considering the specific layout of the actuation system depicted in Fig. 5 (left), the extension of the rear suspension of the reference chord is approximated as

$$\Delta l = \frac{1}{2} l_d = \frac{1 - u_p}{2} l_{d,\mathrm{max}}, \tag{6}$$

where $l_d$ is the deployed length of the depower tape with the maximum value $l_{d,\mathrm{max}} = 1.7$ m. Because we employ a pulley system to decrease the required forces in the actuation system, only half of the length of the depower tape is translated into lengthening or shortening the rear suspension of the reference chord. Equation (6) shows that a full depowering of the wing with $u_p = 0$ leads to a maximum extension $\Delta l_{\mathrm{max}} = 1/2 l_{d,\mathrm{max}}$, from which a maximum depower angle of $\alpha_{d,\mathrm{max}} = 24°$ can be calculated on the basis of Eq. (5).

Aside of a general increase of the aerodynamic load, an increasing angle of attack leads also to a gradual backwards shift of the load distribution, towards the rear of the wing. To balance this load shift, the entire kite has to pitch down, around the bridle point, which increases the angle $\lambda_0$. This aerodynamic characteristic of LEI tube kites has been observed experimentally by Hummel (2017) and van Reijen (2018). Because the chordwise location of the center of pressure controls how the total aerodynamic load is distributed on the front and rear bridle line systems, measuring the line forces is a way to quantify the load shift. To describe how much of the total load is transferred through the front bridle lines, we define the force ratios $F_{t,f}/F_{t,r}$ and $F_{t,f}/(F_{t,f} + F_{t,r})$, where $F_{t,f}$ and $F_{t,r}$ are the magnitudes of the resultant forces transferred through the front and rear bridle line systems, respectively (see also Fig. 11). The force ratios measured for a LEI Hydra V5 kite are illustrated in Fig. 7, indicating

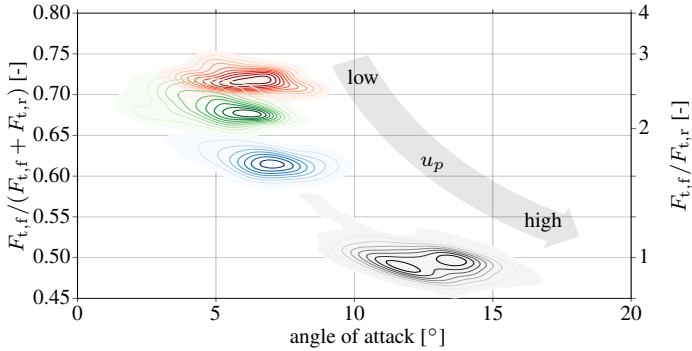

**Figure 7.** Distribution of tensile forces in the bridle line system, from low to high power setting (red, green, blue, black) measured for a commercially available LEI Hydra V5 kite with 14 m$^2$ wing surface area, by Genetrix Kiteboarding (adapted from van Reijen, 2018).

that the aerodynamic load gradually shifts towards the rear bridle lines for increasing power setting $u_{\mathrm{p}}$. At the highest power setting the loads transferred through the front and rear bridle line systems are about equal. Since we did not measure the bridle line forces of the LEI V3 kite we assume a constant position of the center of pressure, derived as an average of several different types of kites by Hummel (2017). Measuring the bridle line force ratio for the LEI V3 kite in flight would help increasing the

5 accuracy of this study but would require additional instrumentation and is recommended for future tests.

However, it is not only the shifting center of pressure that affects the orientation of the kite with respect to the tether. Another important factor is the gravitational and inertial force of the KCU, which contributes almost 40% of the kite system mass and is suspended below at considerable distance from the wing (see Fig. 5). When the kite is flying upwards, the gravitational force is pulling the suspended KCU down, increasing $\lambda_0$, while when it is flying downwards, the effect is inverted and $\lambda_0$ is decreased.

10 When the kite is flying sideways, the mass of the KCU affects the roll orientation of the kite with respect to the tether. In general, the gravitational effect of the KCU increases towards lower elevation angles and lower tension in the tether.

The competing effects of kite aerodynamics and KCU mass are illustrated in Fig. 8 for two extreme load cases. The partially

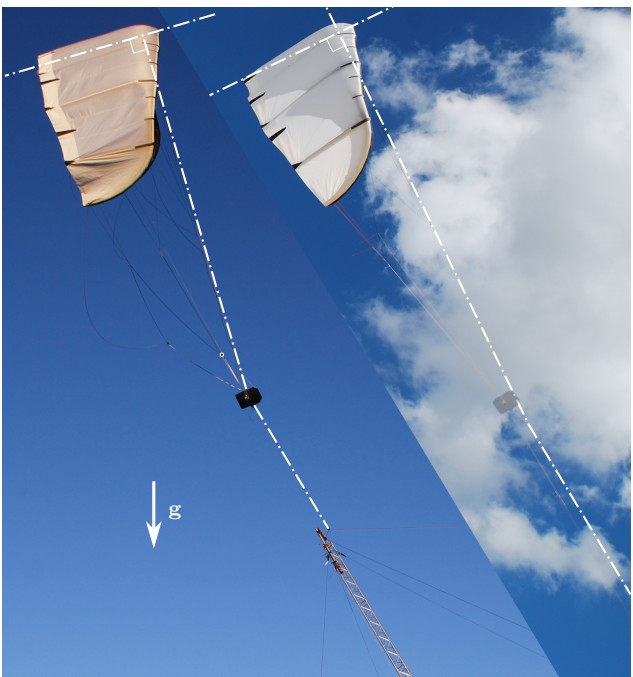

**Figure 8.** Sideview of kite, partially depowered during landing maneuver (left) and fully powered during crosswind flight maneuver (right). The photo on the right was taken during a flight test in which the KCU was replaced by a ring that collected the joined power lines and the two steering lines and redirect them as a triplet of parallel lines to the pilot on the ground. The position of this ring is hinted by an overlaid transparent image of the KCU.

depowered kite on the left is flying statically and is thus only lightly loaded. For this reason, the rear bridle lines are sagging and the wing membrane is not taut. From the photo we can measure a depower angle $\alpha_{\mathrm{d}} = 5.6°$ and a line angle $\lambda_0 = 14.7°$.

The relatively large line angle indicates that the gravitational effect of the KCU mass by far outweighs the aerodynamic effect. On the other hand, the fully powered kite on the right is flying fast crosswind maneuvers and is thus heavily loaded. As a result, the wing membrane and bridle lines are taut. In this particular test, the wing is operated without KCU and $\lambda_0$ depends thus solely on the aerodynamic load distribution on the wing. From the photo we can measure a line angle $\lambda_0 = 5.1°$ and can further recognize that the concept of a reference chord that is perpendicular to the front bridle lines is a good representation of the actual center chord of the wing.

In this work, the wing is idealized as lifting surface with fixed geometry. The proposed geometric depower model is a simplified two-dimensional approximation of the complex three-dimensional aeroelastic response of the bridled membrane wing. The photographic footage depicted in Fig. 9 illustrates how the wing shape changes when transitioning from depowered to powered state. The GoPro® video camera with ultra-wide angle "fisheye" lens captures the entire wing and bridle line

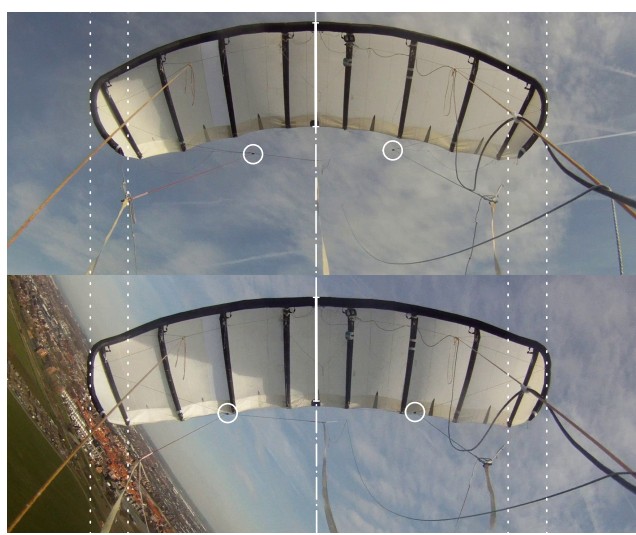

**Figure 9.** Depowered kite (top) and powered kite (bottom) from a video camera mounted on the KCU and looking into the wing. The video sequence of the entire maneuver is available from Schmehl and Oehler (2018).

system, from which we can make several qualitative comparisons. It is obvious that the powering of the wing tensions the entire bridle line system such that the two pulleys (marked by circles) move forward, towards the leading edge. The increasing projected center chord indicates that the wing pitches into the projection plane. The slightly increasing projected span indicates that the entire wing straightens under the substantially increased aerodynamic loading when being powered. This effect is also described by van Reijen (2018, p. 61). Also the curvature (sweep) of the leading edge tube slightly decreases. It is clear that these effects can not be described by a geometric model without accounting for the fluid-structure interaction problem, including membrane wing, bridle line system and steering actuation.

### 3.3 Determining the Lift-to-drag ratio

A common method to estimate the lift-to-drag ratio of a kite is to measure the elevation angle $\beta$ of the tether with the horizontal during static flight (Stevenson, 2003). A disadvantage of this method are the uncertainties arising from the tether sag and the usually unknown wind conditions at the position of the kite. Stevenson (2003) introduces the tether angle of attack $\alpha_{\text{t}}$ to account for all forces acting on the kite system above the bridle point, in our case, the KCU, the bridle line system and the wing. This angle, which is related to the measured inflow angle $\alpha_{\text{m}}$ by Eq. (3), can thus be used to characterize the aerodynamics of the entire kite. However, the value of $\alpha_{\text{t}}$ depends also on the gravitational and inertial forces acting on the kite components. These vary with the specific flight situation such as flying upwards, downwards, sideways or turning maneuvers, as outlined in the previous section.

To understand how the aerodynamic characteristics of the kite are related to the kite design and measured properties we first neglect the effect of gravity. For steady flight, the resultant aerodynamic force $\mathbf{F}_{\text{a}}$ is in equilibrium with the tether force $\mathbf{F}_{\text{t}}$. Because the flexible tether can only support a tensile force but no bending moment, the two forces are tangential to the tether at the bridle point, pointing in opposite directions. The aerodynamic force can be further decomposed into lift and drag components, $\mathbf{L}$ and $\mathbf{D}$, respectively. By definition, the drag force is aligned with the apparent wind velocity vector $\mathbf{v}_{\text{a}}$ and because $\mathbf{F}_{\text{a}}$ is aligned with $z_{\text{t}}$, the lift-to-drag ratio $L/D$ is related to the tether angle of attack $\alpha_{\text{t}}$ by

$$\frac{L}{D} = \cot \alpha_{\text{t}}. \tag{7}$$

When flying on a curved path, as, for example, during figure-of-eight maneuvers, the centrifugal force perpendicular to the tether needs to be balanced by an additional lateral component of the aerodynamic force vector. How this side force $F_{\text{a,s}}$ in $y_{\text{t}}$-direction is generated depends on the specific type of wing and the implemented steering mechanism. Classical rigid wing concepts with aerodynamic control surfaces (Ruiterkamp and Sieberling, 2013) and the Skysails ram air wing (Erhard and Strauch, 2013) roll the wing such that the lift vector tilts towards the center of turn. Most flexible membrane wing concepts, on the other hand, yaw and twist the wing, using the vertical surface of the wing tips to generate a side force and turning moment. This mechanism is depicted in Fig. 4 and described in more detail in Bosch et al. (2013, Sect. 17.3.1) and Fechner and Schmehl (2018, Sect. 15.2.2).

In a similar way, the effect of gravity needs to be balanced by an additional component of the aerodynamic force vector. This is formally expressed by the force equilibrium at the bridle point for steady flight

$$\mathbf{F}_{\text{a}} + m\mathbf{g} + \mathbf{F}_{\text{t}} = 0. \tag{8}$$

However, in difference to the centrifugal acceleration during turning maneuvers, the resultant gravitational force $m\mathbf{g}$ acts not only sideways but depending on the orientation of the kite in all three directions, $x_{\text{t}}$, $y_{\text{t}}$ and $z_{\text{t}}$. To derive the required balancing components of $\mathbf{F}_{\text{a}}$ we express the resultant gravitational force of all kite components in the tether reference frame

$$m\mathbf{g} = \begin{bmatrix} -\cos\beta\cos\psi \\ \cos\beta\sin\psi \\ \sin\beta \end{bmatrix} mg. \tag{9}$$

This representation is based on the assumption of a straight tether, such that the angle between the horizontal and the $z_t$-axis is identical to the elevation angle $\beta$ of the kite (see Fig. 6).

The force equilibrium given by Eq. (8) is illustrated in Fig. 10 for the special case of an upwards oriented kite ($\psi = 0°$). The

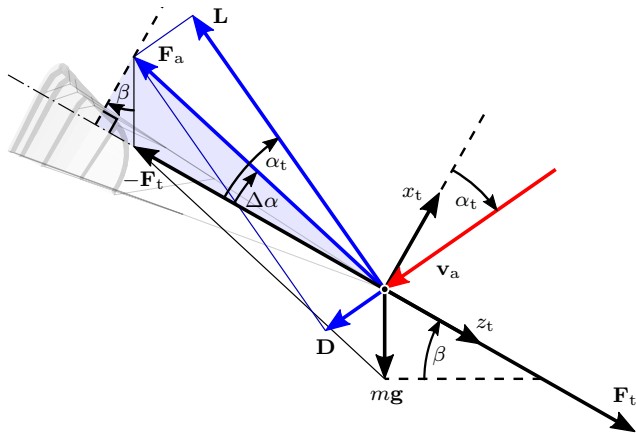

**Figure 10.** Force equilibrium of a kite in steady-state flight, for the special case of the kite oriented upwards with $\psi = 0°$, flying in the plane spanned by the wind velocity vector and the vertical, described by $\phi = 0°$. The forces acting on the kite components are lumped to the bridle point. See (Schmehl et al., 2013, Fig. 2.11) for an illustration of the force equilibrium extended to the general case of kite flight in three dimensions.

vector diagram shows how the gravitational force is compensated by an upwards rotation of the aerodynamic force by an angle $\Delta\alpha$. For arbitrary orientation of the kite, the aerodynamic force components that are required to compensate the gravitational force are given by the inverse of Eq. (9). Considering the compensation in the $x_t z_t$-plane only, we can derive the following relation between tether force, gravitational force and the compensation angle $\Delta\alpha$

$$\tan(\Delta\alpha) = \frac{mg\cos\beta\cos\psi}{F_t + mg\sin\beta}, \tag{10}$$

which is illustrated by the shaded right triangle in Fig. 10 for the special case $\psi = 0°$. Using the tether angle of attack $\alpha_t$ defined by Eq. (3), the lift-to-drag ratio can be determined from

$$\frac{L}{D} = \cot(\alpha_t - \Delta\alpha). \tag{11}$$

The gravitational force in $y_t$-direction is compensated by a steering force

$$F_{a,s} = -mg\cos\beta\sin\psi, \tag{12}$$

which, for the investigated type of kite, is generated by a sideslip angle $\beta_s$ (Fechner and Schmehl, 2018). When flying figure-of-eight maneuvers, the angles $\beta$ and $\psi$ are continuously varying and the gravity compensation is accordingly alternating through $x_t$-, $y_t$- and $z_t$-directions. Neglecting this effect would have the consequence that the measured aerodynamic characteristics seemingly vary along the flight maneuver.

For orientations with upward component ($-90° < \psi < 90°$), we obtain positive values for $\Delta\alpha$. For orientations with downward component, gravity opposes the aerodynamic drag of the wing resulting in negative values for $\Delta\alpha$. The elevation angle $\beta$ of the kite is determined by the position of the kite with respect to the ground station (see Fig. 6) and only in case of a straight tether identical to the inclination of the tether force (see Fig. 10). One of the key advantages of the described measurement method is that sagging of the tether does not directly affect the measurement of $L/D$. We use the elevation angle $\beta$ only to correct for the effect of gravity in Eq. (10). This correction is also affected by the ratio of gravitational force to tether force. In contrast to this, sagging has a direct effect for methods that are based on ground-based measurements of the tether angle of attack, as proposed, for example, by Hummel (2017).

The tether angle of attack $\alpha_t$ can be calculated from Eq. (3), using the measured inflow angle $\alpha_m$ and the line angle $\lambda_0$. The latter is determined numerically by solving for the quasi-steady force equilibrium of the simplified mechanical model illustrated in Fig. 11. In this framework, the individual components of the kite are idealized as point masses which are exposed

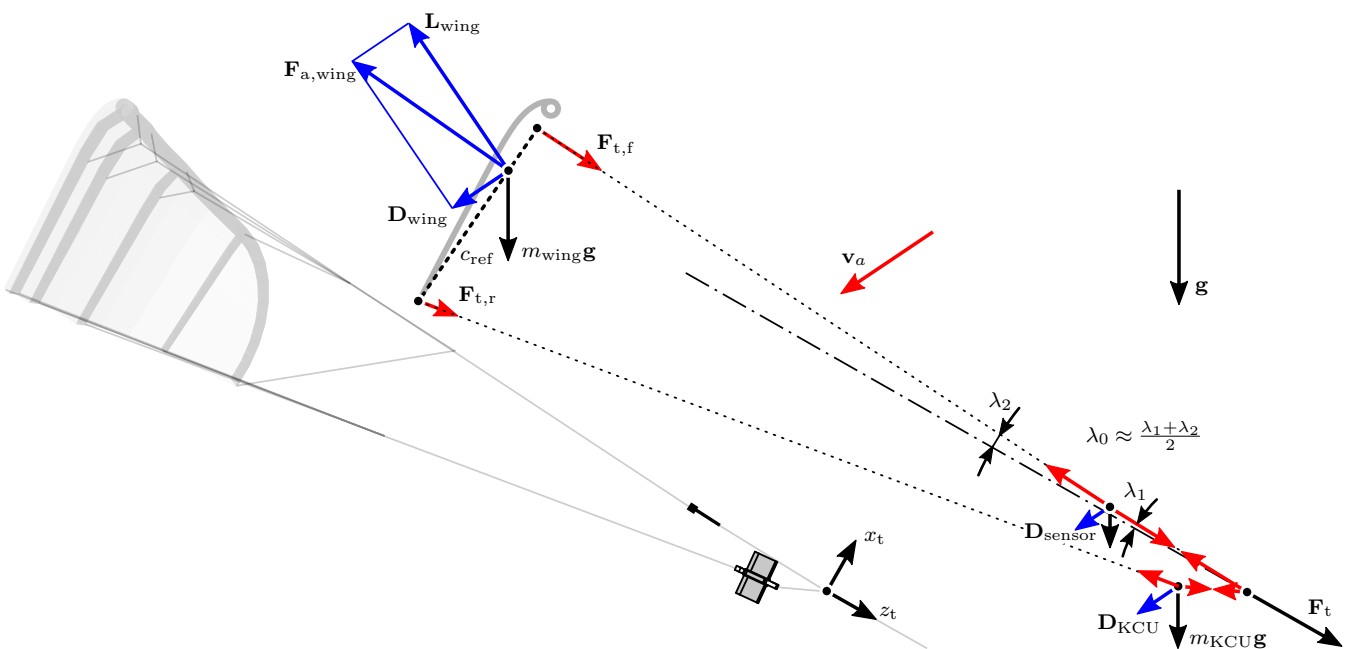

**Figure 11.** Fully powered kite (left) and simplified mechanical model of the kite system (right), including wing, measurement setup and KCU, showing external forces (black: gravitational forces and tether force at the bridle point, blue: aerodynamic forces) and internal forces (red: bridle line forces) to calculate the bridle line angles $\lambda_1$ and $\lambda_2$. Depicted is the special case of an upwards oriented kite with apparent flow velocity and all model forces in the drawing plane. Force vectors are not to scale.

to external forces (gravity, aerodynamic lift and drag, tether force at the bridle point) and internal bridle line forces. The drag and the mass of the bridle line system are assigned to the wing. The total resultant aerodynamic force and the gravitational

force acting on the kite components are thus decomposed as

$$\mathbf{F}_a = \mathbf{L}_{\text{wing}} + \mathbf{D}_{\text{wing}} + \mathbf{D}_{\text{KCU}} + \mathbf{D}_{\text{sensor}}, \tag{13}$$

$$m\mathbf{g} = (m_{\text{wing}} + m_{\text{KCU}} + m_{\text{sensor}})\,\mathbf{g}. \tag{14}$$

In a first step, we calculate the resultant aerodynamic force $\mathbf{F}_{\text{a,wing}} = \mathbf{L}_{\text{a,wing}} + \mathbf{D}_{\text{a,wing}}$ that is required to balance the given tether force $\mathbf{F}_t$, and the aerodynamic and gravitational forces acting on the kite components. Approximating the KCU and the measurement setup as blunt bodies with an aerodynamic drag coefficient of $C_D = 1.0$, we calculate a drag contribution of the KCU of about 10% of the wing drag and a contribution of the measurement setup of about 1%.

In a second step, we use a shooting method to iteratively adjust the bridle line angles until the two-dimensional model geometry for the known external forces and bridle line lengths is in quasi-steady equilibrium. For this we assume that a bridle line force is always in line with the connection line of the two attachment points. We further assume that the center of pressure and the center of mass of the wing are both at 25% of the reference chord (see Fig. 11). This is in line with the mass distribution used by Bosch et al. (2013) and the average ratio of 3:1 for the forces in front and rear bridle lines measured by Hummel (2017) for different kites at various power settings. Starting from an initial guess for the line angle $\lambda_1$, we calculate the angle $\lambda_2$ and the respective angles for the steering lines. Based on the resulting geometry, we then compute the distance between the front and rear bridle attachments on the chordline of the wing. If this distance is larger than $c_{\text{ref}}$, the value of $\lambda_1$ is reduced and the calculation repeated. The iteration loop is terminated when the target distance $c_{\text{ref}}$ of the bridle attachment points is reached. The algorithm generally converges within 4 iterations, using a termination criterion of 0.01 m or 0.5%.

Compared to ground-based methods, for example, with angular sensors at the ground attachment point of the tether, the sagging of the tether does not affect the measurement significantly. Also, the effect of gravity on the measurement setup was found to be negligible. This is because the measurement setup is a lightweight construction compared to the KCU and because the power lines are generally well tensioned. We have observed that the line angles $\lambda_1$ and $\lambda_2$ in general differ only by $0.1°$ to $0.2°$, such that the power lines can practically be considered straight. We thus use the mean value of $\lambda_1$ and $\lambda_2$ as line angle $\lambda_0$. The KCU, on the other hand, has a considerable effect, especially during reel-in maneuvers when the force in the rear suspension lines is of the order of the gravitational force of the KCU.

The calculated values vary between $0° < \lambda_0 < 2°$ for flying downward. For upward flight and during reel-in we find values of $3° < \lambda_0 < 7°$. For low tether tension and upward flight values of $10° < \lambda_0 < 12°$ occur. These computed ranges agree well with photographic evidence, such as the snapshots shown in Fig. 8. The highest values occur when both tether tension and elevation angle are low which is the case during launch and landing.

When all lines are well tensioned and straight, the pitching of the kite around the bridle point does not affect the bridle geometry. However, the rear bridle lines are not always well tensioned, as can be seen clearly for the landing maneuver shown in Fig. 8 (left). When flying upwards during power production (see Fig 11), the effects of drag and gravity are both in a downward direction which can cause a measurable sag of the rear bridle lines. This effective shortening of the bridle lines increases the powering of the kite and can be modeled as a reduction of the depower angle $\alpha_d$.

### 3.4 Determining the Lift coefficient

The lift coefficient $C_L$ is a dimensionless number,

$$L = \frac{1}{2}\rho C_L v_a^2 A, \tag{15}$$

characterizing the lift force as a function of the air density $\rho$, relative flow velocity $v_a$ and projected wing surface area $A$. Density and relative flow velocity are measured directly, while a constant value for the projected wing surface area is used (see the table included in Fig. 5). Using the lift-to-drag ratio $L/D$ we can compute the lift force generated by the kite as

$$L = F_a \sqrt{\frac{\left(\frac{L}{D}\right)^2}{1+\left(\frac{L}{D}\right)^2}}. \tag{16}$$

We resolve Eq. (8) in horizontal and vertical directions to relate the force magnitudes as

$$F_a = \sqrt{(F_t \cos\beta)^2 + (F_t \sin\beta + mg)^2}, \tag{17}$$

again making use of the idealization that the tether force is aligned with the radial direction from the ground attachment point. The special case of an upwards oriented kite with all forces in the drawing plane is illustrated in Fig. 10.

## 4 Results

The data for this study was acquired during a one hour test flight of the prototype described in Sects. 2 and 3.2 on 24 March 2017 at the former naval airbase Valkenburg, close to Leiden, the Netherlands. A video camera mounted on the measurement setup documented that all sensors were moving freely in the airflow and did not exhibit any visible faulty behavior. This is illustrated in Fig. 12, with the diagram showing 27 seconds at the beginning of a representative traction phase.[1] The first 180

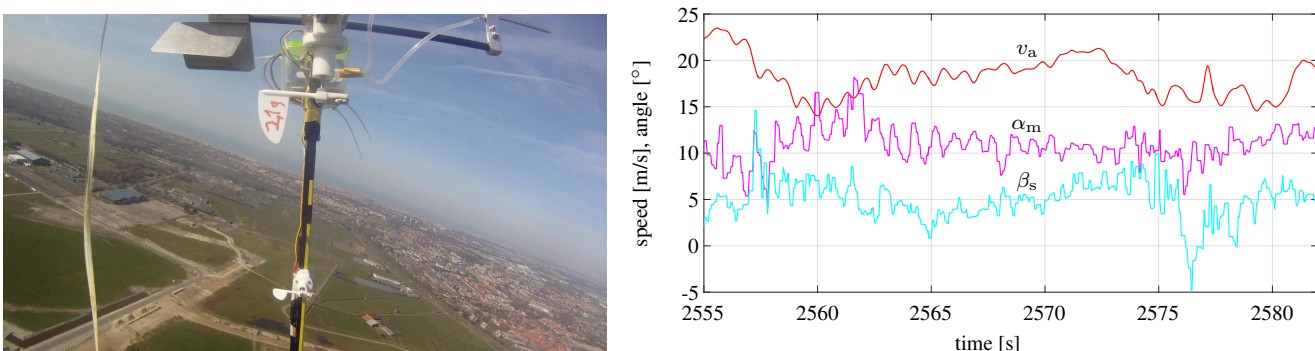

**Figure 12.** Video still of the relative measurement setup taken from the right power line (left), raw values of apparent flow velocity $v_a$ and inflow angles $\alpha_m$ and $\beta_s$ recorded over time at the beginning of a representative traction phase (right).

[1]In this study, time is counted from the launch of the kite, starting at $t = 0$.

seconds of the one hour test flight are available as video footage from Oehler and Schmehl (2018). The maximum apparent flow velocity occurs during the first two seconds of the depicted time window, when the kite transitions from the retraction to the traction phase. Because the kite flies downwards during this maneuver, it is additionally accelerated by the effect of gravity, which leads to a temporary increase of the apparent flow velocity.

Our measurements contradict the earlier study of Ruppert (2012), who reported considerable variations of the angle of attack (up to 30° during the traction phase) and sideslip angle ($-20° < \beta_\mathrm{s} < 20°$). In our study, the angle of attack is limited to a narrow range of $6° < \alpha < 16°$ during the traction phase. The measured sideslip angle deviates from its mean value by a maximum of $\Delta\beta_\mathrm{s} = 10°$ only during very sharp turns, which is indicative for the high aerodynamic side force produced by a moderate side slip angle. We conclude that an accurate determination of the relative flow at the kite is not feasible without in situ measurements at the kite. Using only GPS and IMU data and ground based measurements, as proposed by Ruppert (2012), leads to a substantial degradation of the achievable accuracy. The apparent flow speed is around $v_\mathrm{a} = 18$ m/s during the traction phase and $v_\mathrm{a} < 15$ m/s during the retraction phase. In the analyzed data set, the mean value of the sideslip angle was not zero, which we would have expected for a symmetric kite. This offset resulted from an asymmetric layout of the bridle lines, causing the kite to fly in an asymmetric pattern during the traction phase. We recommend to investigate the effect of the sideslip angle on the kite aerodynamics in a future study, using alternative data for a verified symmetric layout of the bridle line system.

A common technique to analyze measurement data from wind turbines or other rotating machinery is phase averaging. In contrast to Behrel et al. (2018) we did not use this technique because of the difficulty to determine a clear phase location of the data. Harvesting wind energy with tethered flying devices operated in pumping cycles has many more degrees of freedom than conventional wind turbines and even though the operation in a variable wind environment requires these to be actively controlled, the location of the lightweight devices along the flight path is tightly coupled to the evolution of the wind field along this path. For a wind turbine, with rotor blades that are mechanically linked and have a comparatively large rotational inertia, the determination of a phase location is comparatively straightforward. Instead of using rigorous phase averaging, we only differentiate between traction and retraction phases, subdividing the crosswind maneuvers further into flying upwards (against gravity) and flying downwards (with gravity). This can be regarded as a low resolution phase averaging, tailored to the specific physics of tethered flight in pumping cycles. However, the available data covered only five separate cycles, which is by far not sufficient for a meaningful statistical analysis.

### 4.1 Reeling oscillations

The flight data illustrated in Fig. 12 exhibits strong fluctuations at a distinct frequency of 1.2 Hz in both $v_\mathrm{a}$ and $\alpha_\mathrm{m}$. These oscillations occur repeatedly for several seconds during the retraction and traction phases. Other independently measured variables also exhibit this behavior, for example, the tether force $F_\mathrm{t}$, the tether reeling speed $v_\mathrm{t}$, the pitch rate of the kite, as well as the forward and downward accelerations measured by the wing-mounted IMU. To identify the cause of these oscillations we considered two possible mechanisms in a previous study (Oehler and Schmehl, 2017): a first mode of radial oscillations of the kite that are commanded by the reeling control of the ground station and a second, flight dynamic mode.

These tangential oscillations in forward/backward direction are kinematically coupled to pitch oscillations. Based on a simple model of a driven oscillator, we determined for the flight dynamic mode a relatively strong damping, with a coefficient $\zeta_k$ of 0.63, and eigenfrequencies $f_k$ of 0.81 Hz for the traction phase and 0.39 Hz for the retraction phase. Because these values differ from the frequency of the observed fluctuations we conclude that we are not observing a flight dynamic mode of the kite system but that the reeling controller of the ground station is the root cause of the oscillations. This is supported by the additional observation that the fluctuations cease when the reeling of the tether stops. It is clear that this behavior could be suppressed by an adjustment of the ground station controller, however, this is not part of the study.

To estimate the effect of these forced oscillations with frequency $f_{GS} = 1.2$ Hz on the kite aerodynamics we determine the reduced frequency (Hassig, 1971)

$$k = \frac{f \pi c}{v_a}. \tag{18}$$

Using a chord length of $c = 2.7$ m and an apparent flow speed of $v_a = 20$ m/s, we calculate a value of $k_{GS} = 0.5$. This means that the flow around the kite is unsteady, which in turn can cause a phase shift of the aerodynamic load with respect to the angle of attack. To mitigate the effect of a possible phase shift, we smoothen the data over an interval of $T = 2.5$ s which is equivalent to three oscillation periods. In doing this we essentially regard the forced oscillations and resulting unsteady aerodynamics of the kite as a subscale process, which we filter out to retain the assumption of quasi-steady flight.

To assess the effect of the turning maneuvers during crosswind flight on the aerodynamics of the kite we determine a characteristic frequency of $f_{turn} = 0.1$ Hz which corresponds to a half turn in about five seconds. The corresponding reduced frequency of $k_{turn} = 0.042$ indicates that the aerodynamic time scale is more than an order of magnitude smaller than the turning time scale, which confirms the assumption of quasi-steady flight also from this perspective.

## 4.2  Lift-to-drag ratio

The lift-to-drag ratio $L/D$ is a key parameter to characterize the aerodynamic performance of a wing. As described by Eq. (1), this parameter determines how fast a kite can theoretically fly in a given wind environment and by that also what tether force can be achieved for a given size of the wing (Loyd, 1980; Schmehl et al., 2013). In contrast to a conventional aircraft, the C-shaped flexible membrane wing is used as a single aerodynamic control surface with the double function of steering and generating a tether force that can be modulated over a wide range. This is of particular importance for the considered operation in pumping cycles because the achievable net energy per cycle crucially depends on the ability of the wing to alternate between a high lift-to-drag ratio during the traction phase and a low ratio during the retraction phase.

In Fig. 13 we investigate the influence of the angle of attack $\alpha$ of the wing and the power setting $u_p$ of the kite. The lift-to-drag ratio of the entire kite (all components from bridle point outwards) is derived on the basis of Eq. (11), using Eq. (3) to account for the measured flow angle $\alpha_m$ and the estimated line angle $\lambda_0$, and Eq. (10) to account for the effect of gravity, expressed as compensation angle $\Delta\alpha$. No further filtering or smoothing is applied to the data. Although the effect of gravity on all kite components is taken into account as well as the aerodynamic drag on KCU and measurement setup, the data is still

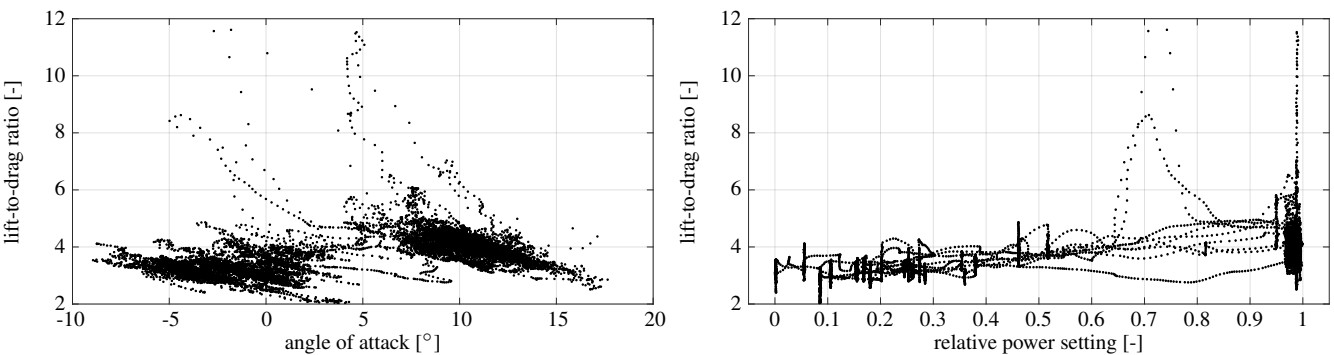

**Figure 13.** Measured lift-to-drag ratio $L/D$ of the kite plotted over the angle of attack $\alpha$ of the wing (left) and relative power setting $u_{\mathrm{p}}$ of the kite (right). No filtering or smoothing is applied to the data.

scattered considerably. In the following we will show that this is for a considerable degree due to occasional dips in the tether tension, steering actuation and the associated sideslip angle.

In Fig. 13 (left) we can distinguish a distinct region of lower angle of attack, $-7° < \alpha < 3°$, indicating the retraction phases, and a distinct region of higher angle of attack, $7° < \alpha < 15°$, indicating the traction phases. In Fig. 13 (right) the retraction

phases are indicated by power settings $u_{\mathrm{p}} < 0.55$, while the traction phases are indicated by power settings $u_{\mathrm{p}} \approx 1$. Values in between these regions are typical for the transition between the retraction and traction phases. During the traction phases we measure an average lift-to-drag ratio of about $L/D = 4$, during the retraction phases we measure an average ratio of about $L/D = 3$, which is desired to reduce the tether force and thus also the energy consumption during retraction of the kite.

In the next step we filter the data as outlined in Table 2, reducing the spreading and removing outliers. The correlated effect

**Table 2.** Three filtering procedures applied to the measured lift-to-drag ratio.

| filter | description | reason | visible effect |
|---|---|---|---|
| 1 | moving average over $T = 2.5$ s | forced oscillations with $f_{\mathrm{GS}}$; remove subscale dynamics. | reduces spread during retraction |
| 2 | $F_{\mathrm{t}} > 400$ N | model limitation | eliminate outliers |
| 3 | exclude steering | strong deformation | eliminate outliers |

of the angle of attack and the relative power setting on the lift-to-drag ratio is illustrated in Fig. 14, where we have also applied the moving average smoothing described in Sect. 4.1 (filter #1). To identify the cause of the high $L/D$ values, we further exclude data points with a tether force $F_{\mathrm{t}} < 400$ N (filter #2). For such low tether tensions the assumptions of a straight tether and quasi-steady flight state are not valid anymore, which can lead to substantial measurement errors. Excluding data points with $F_{\mathrm{t}} < 400$ N in fact eliminates many of the unphysically high $L/D$ values.

The diagrams of Figs. 13 and 14 show that for an increasing power setting $u_{\mathrm{p}}$ the angle of attack $\alpha$ and also the lift-to-drag ratio $L/D$ increases. A low angle of attack results in a low lift force and therefore a low force ratio. The maximum of

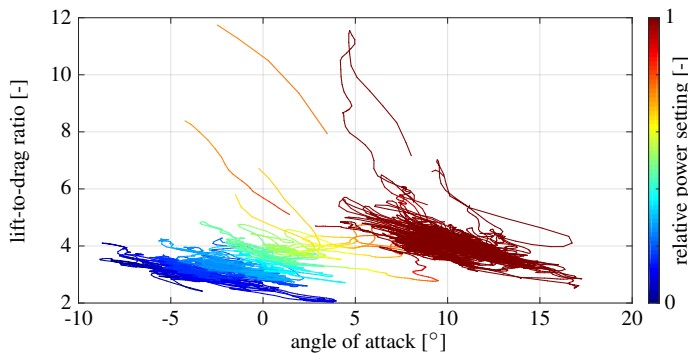

**Figure 14.** Measured lift-to-drag ratio $L/D$ of the kite plotted over the angle of attack $\alpha$ and colored by the relative power setting $u_\mathrm{p}$. The coloring ranges from blue, for lower values of $u_\mathrm{p}$ when retracting the kite, up to dark red, for the fully powered kite with $u_\mathrm{p} = 1$ during the traction phase. Table 2 filters #1 and #2 have been applied.

about $L/D = 5$ occurs in the range $5° < \alpha < 10°$ and is only reached for $u_\mathrm{p} = 1$. For higher angle of attack the force ratio decreases again because of the substantially increasing drag force. The measured dependency follows the same general trend as for conventional aircraft wings and was already observed by van der Vlugt et al. (2013).

Figure 15 shows the temporal evolution of $u_\mathrm{p}$ and $L/D$ during pumping cycle operation. During the traction phases with

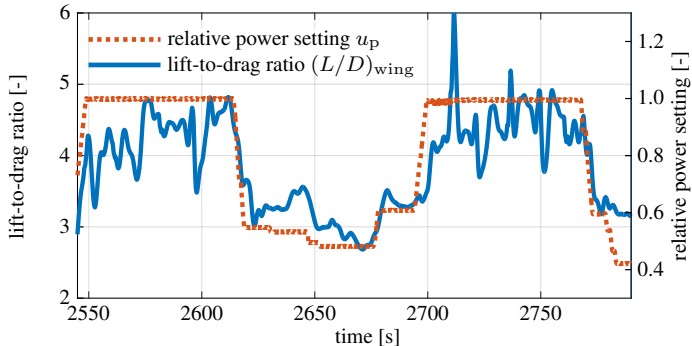

**Figure 15.** Evolution of the lift-to-drag ratio during pumping cycle operation.

5  $u_\mathrm{p} = 1$ we observe periodic drops to force ratios $L/D < 4$. The drops are correlated with the turning maneuvers and are caused by the steering-induced deformation of the wing and the additional drag component of the required side force (Fechner et al., 2015). To investigate the effect of the steering on the entire data set, we color in Fig. 16 the measured $L/D$ data by the steering intensity. We can recognize that very strong turning maneuvers coincide with a low tether force and extreme force ratios. During the traction and transition phases, the lift-to-drag ratio for a specific power setting is significantly lower when

10  the steering system is active. This has been shown also in Oehler et al. (2018). Next to the described effects of deformation and steering-induced drag, there is also a feedback loop because an increasing drag lowers $L/D$ which in turn increases $\alpha_\mathrm{t}$ and to a certain extent also $\alpha$. The increasing $\alpha$ lowers $L/D$ further.

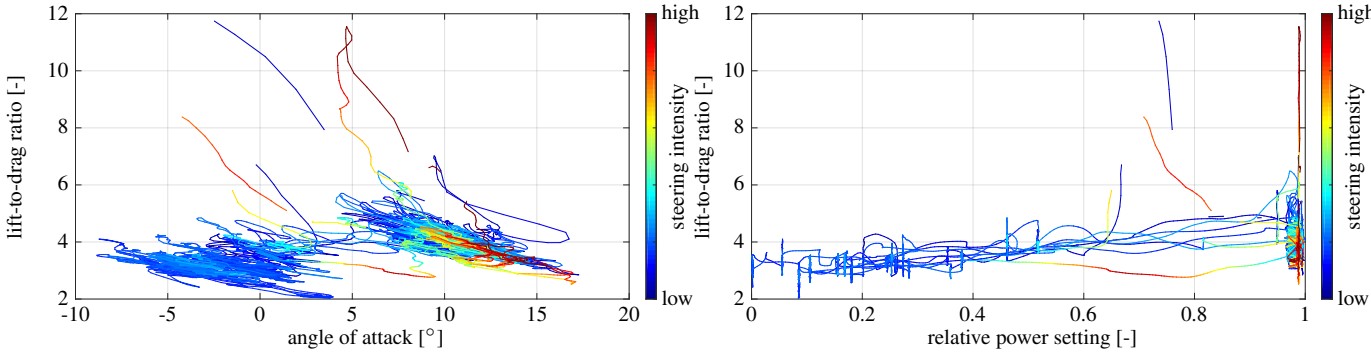

**Figure 16.** Measured lift-to-drag ratio $L/D$ of the kite plotted over the angle of attack $\alpha$ of the wing and colored by the relative power setting $u_{\mathrm{p}}$, colored by the steering intensity, ranging from blue, for no steering, up to yellow and red, for strong steering actuation during turning maneuvers. Table 2 filters #1 and #2 have been applied.

### 4.3 Comparison with aerodynamic models

Ruppert (2012) and Fechner et al. (2015) present two different real-time-capable models for the dynamic simulation of pumping kite power systems. In both approaches the aerodynamics of the kite is described by $C_{\mathrm{L}}(\alpha)$ and $C_{\mathrm{D}}(\alpha)$ correlations that have been derived from existing measurement data of two-dimensional sail wing sections. According to the authors, major adjustments were required to fit the simulated flight behavior of the kite to measured reference trajectories. Both dynamic models predict the flight path and power production with reasonable accuracy for a broad range of operational conditions and are thus suitable for optimization of kite control.

 Two different definitions of the angle of attack are used. Fechner et al. (2015) measure the angle from the center chord to the relative flow velocity vector, while Ruppert (2012) measures it from the orientation of the wing-mounted IMU. Both definitions are difficult to reproduce experimentally for subsequent measurement campaigns because the orientation of the center chord is a virtual geometric property and can only be estimated, while the IMU is mounted on one of the inflatable struts with Velcro® tape which introduces a considerable degree of uncertainty, even when using the same kite.

 To compare the two existing sets of aerodynamic correlations with our measurement data we first need to eliminate the offsets introduced by the different definitions of $\alpha$. For this purpose we shift the $L/D$ correlations of Ruppert (2012) and Fechner et al. (2015) in the $\alpha$-range such that the maxima occur at $\alpha = 7.5°$, which is where the maximum average $L/D$ of our data set is located. For reference we note that the maxima of the unshifted correlations occurred at $\alpha = 12.5°$ (Ruppert, 2012) and $\alpha = 16°$ (Fechner et al., 2015).

 As stated above the lift-to-drag ratio decreases during turning maneuvers because of the additional drag of the wing tips. This can be clearly recognized from the data plotted in Fig. 15, which exhibits strong variations when flying crosswind maneuvers during the traction phases. On the other hand, the existing aerodynamic correlations have been derived for a wing in straight flight, with symmetric steering input. Ruppert (2012), for example, has excluded from his analysis data points that were associated with strong asymmetric steering input. We have applied a similar filtering procedure to our data. In Fig. 17 we

compare the filtered data with the existing aerodynamic correlations. The correlation of Ruppert (2012) is mainly based on five

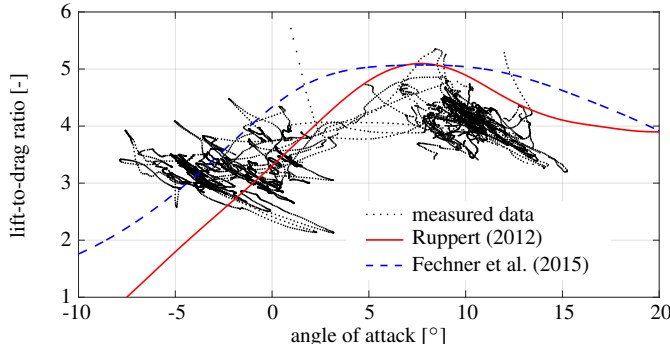

**Figure 17.** Comparison of measured lift-to-drag ratio with existing aerodynamic correlations. Table 2 filters #1, #2 and #3 have been applied.

data sets acquired with a LEI V2 kite with 25 m$^2$ wing surface area, as shown in Fig. 1 (right), and one data set acquired with a smaller LEI Hydra V5 kite, as used for the diagram in Fig. 7. On the other hand, the correlation of Fechner et al. (2015) is based on aerodynamic models for stalled and unstalled airfoils from Spera (2008), with additional experience-based modifications

for achieving a better fit with the aerodynamics of a LEI tube kite. For system level modeling, van der Vlugt et al. (2019) distinguish between a large (LEI V3) and a small (LEI Hydra V5) kite, using lift-to-drag ratios of 3.6 and 4.0, respectively, during the traction phases as opposed to 3.5 and 3.1, respectively, during the retraction phases.

Overall, we find a reasonable agreement between our measured data and existing aerodynamic characterization attempts. The correlations of Ruppert (2012) and Fechner et al. (2015) slightly overestimate the lift-to-drag ratio, with force ratios $L/D > 4$

even for angles $\alpha > 15°$. This is consistent with the common assumption of a high angle of attack during the traction phase (van der Vlugt et al., 2013). Our measurements show, however, that the angle of attack is lower and generally does not exceed $\alpha = 15°$. The lift-to-drag data proposed by van der Vlugt et al. (2019) for traction and retraction phases corresponds very well with the average lift-to-drag ratios measured in these phases. The depowered kite ($u_\mathrm{p} < 0.5$) and the powered kite ($u_\mathrm{p} = 1$) show different trends. The data plotted in Fig. 14 indicates that the lift-to-drag ratio of the depowered wing depends mainly on

the power setting $u_\mathrm{p}$, while the effect of the angle of attack is only minor. In contrast to this, the force ratio of the powered kite depends mainly on the angle of attack with $L/D$ decreasing for increasing $\alpha$.

With Eq. (4) we have formally separated three fundamental contributions to the angle of attack $\alpha$ of the wing. While the tether angle of attack $\alpha_\mathrm{t}$ and the line angle $\lambda_0$ represent the contributions due to flight motion and pitching of the entire kite with respect to the tether, the depower angle $\alpha_\mathrm{d}$, which is linked to the relative power setting $u_\mathrm{p}$ by Eqs. (5) and (6), also causes a

complex deformation of the bridled membrane wing (see Sect. 3.2). The spanwise twisting and bending has a strong secondary effect on the aerodynamics of the wing and accounting only for the dependency on $\alpha$ leads to considerable uncertainty of the measured aerodynamic characteristics. This effects is one of the contributing factors for the broad spreading of the data in Fig. 17. For this reason, we recommend to keep the relative power setting as a separate influencing parameter, next to the angle

of attack, to improve the aerodynamic characterization of a pumping cycle AWES over the whole flight envelope. In fact, the transition from powered to depowered state of the wing should be regarded as a sequence of different wings.

## 4.4 Lift coefficient

The tests considered in this study are based on a constant force control strategy for the traction phases, with a set value $F_{t,o} = 3.25$ kN. Whenever the actual tether force $F_t$ drops below this value, the ground station reduces the reeling speed $v_t$, when the force exceeds this value, it increases the reeling speed. This control strategy ensures that the aerodynamic loading of the system is limited despite operating in a fluctuating and varying wind environment.

In Fig. 18 we plot the measured lift coefficient $C_L$ of the kite as a function of the relative flow velocity $v_a$, colored by the heading. The diagram only includes data from the traction phases and when $F_t > 3$ kN. Flight situations which do not meet

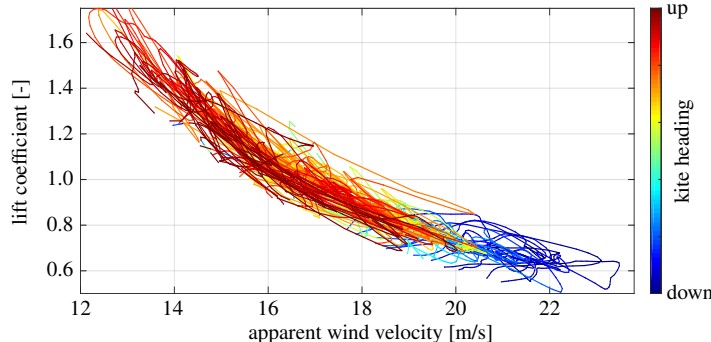

**Figure 18.** Measured lift coefficient $C_L$ of the kite as a function of the relative flow velocity $v_a$ and colored by the heading. The heading range from down to up covers both heading angle ranges $180° > \psi > 0°$ and $180° < \psi < 360°$ equally (see Fig. 6).

this condition are, for example, the transitions to and from the retraction phase or sharp turning maneuvers. Because of Eq. (15) and the constant force control the data points are correlated by

$$C_L v_a^2 = \text{const.} \tag{19}$$

Figure 18 clearly shows how the flight motion of the kite adjusts continuously to the force balance that varies along the crosswind maneuvers to maintain the commanded tether force $F_{t,o}$. As a result of gravity, the kite flies faster, with lower $\alpha$ and $C_L$ on trajectory segments with a downwards component, while it flies slower, with higher $\alpha$ and $C_L$ on segments with an upwards component. Because of the constant force control, the relative flow velocity $v_a$ exhibits an inversely proportional behavior to the angle of attack $\alpha$ and the lift coefficient $C_L$. The inversely proportional correlation of $\alpha$ and $v_a$ can also be recognized in Fig. 12.

Van der Vlugt et al. (2019, Sect. 2.4) show that the angle of attack of a massless kite with constant power setting does not vary along its flight path through a constant uniform wind field. The described effect of gravity and the natural wind environment induce a variation of the angle of attack, although the power setting is kept constant at $u_p = 1$ when flying

crosswind maneuvers. Because of the constant power setting, the wing is not deforming and the variations of $C_L$ and $L/D$ can be attributed solely to changes in of the angle of attack.

Figure 19 shows the measured lift coefficient $C_L$ as a function of the angle of attack, colored by the heading of the kite. To better differentiate the effect of the heading, we subdivide the range from pointing downwards to pointing upwards into 10

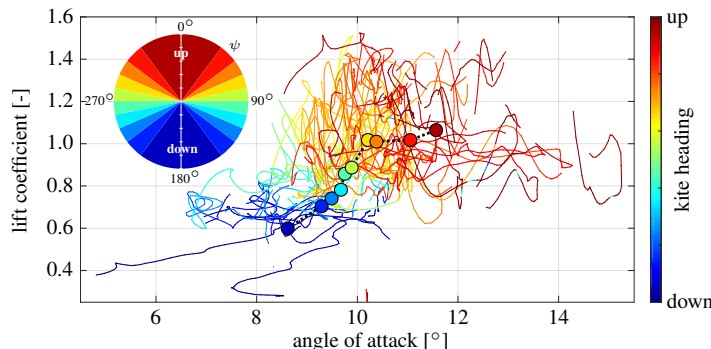

**Figure 19.** Measured lift coefficient $C_L$ of the kite as a function of the angle of attack $\alpha$, colored by the heading.

classes. The heading classes are equidistant in $\cos \psi$, i.e. $\Delta \cos \psi = 0.2$, as indicated by the circular legend in Fig. 19. Per class we compute the average data point and display this as a symbol according to the color legend. The dark blue data point with the lowest angle of attack thus represents the average of all measured flight conditions with a heading that is most closely aligned with the gravity vector.

Despite of the filtering, Figs. 17 and 19 show still a considerable dispersion of the measured data. The various idealizations

required to model the flexible membrane kite system and the assumption of quasi-steady flight with negligible inertial effects contribute to that as well as the fact that the evaluated pumping cycles differed in flight path, wind conditions and many other parameters. Yet, we can recognize two clear trends of the averaged data:

  – the lift coefficient increases with the angle of attack, and

  – the angle of attack and lift coefficient are higher when flying upwards.

The first trend reflects the common aerodynamic characteristics of a wing, while the second trend is caused by the constant force control strategy and was already observed by Oehler (2017). The average lift coefficient plotted in Fig. 19 exhibits a steep slope for lower angles of attack. At $\alpha = 9°$ we measure an average value $C_L = 0.7$ while at $\alpha = 12.5°$ this value has risen to $C_L = 1.0$, which is close to the ideal case of a two-dimensional lifting surface. For wings with low aspect ratio, such as the considered soft kite, we generally expect a more gentle slope of the lift coefficient. The increasing camber and flattening of the

wing for higher angles of attack are two mechanisms that can contribute to this steep slope (see Sect. 3.2 and Fig. 9). Since we use a constant reference wing surface area in Eq. (15) these mechanisms increase the lift coefficient.

Because the power setting of the kite is kept constant during the traction phase we can not actively control the angle of attack of the wing. Instead, the angle results from the quasi-steady force equilibrium of the kite and is thus affected by the

varying gravitational force contribution and the wind environment. Our analysis shows that for operation with constant force control the heading of the kite has the strongest influence on the angle of attack during crosswind maneuvers. When the kite is flying upwards, drag and gravitational force are pointing in similar directions, while for downwards flight, both forces point in opposite directions. This causes the differences in relative flow velocity in Fig. 18 and in angle of attack in Fig. 19.

## 5   Conclusions

In this study we present a method to determine the lift-to-drag ratio and lift coefficient of a soft kite during flight operation by in situ measurement of the relative flow. Tailored towards a kite system with suspended control unit, the flow sensor is installed in the power lines and independently measures the magnitude of the relative flow velocity, the sideslip angle and an orthogonal inflow angle from which the angle of attack of the wing is derived. The effect of gravity on the individual kite components is taken into account in processing the data as well as the aerodynamic drag of kite control unit and measurement setup. Further included are a smoothening procedure to remove the effect of low frequency oscillations induced by the ground station, and filtering procedures to remove the effects of too low tether tension and high steering intensity.

We distinguish three fundamental contributions to the angle of attack of the wing: the tether angle of attack $\alpha_\mathrm{t}$, which is related to the flight motion of the kite, the line angle $\lambda_0$, which characterizes the pitch of the entire kite relative to the tether, and the depower angle $\alpha_\mathrm{d}$, which characterizes the pitch of the wing relative to the kite due to depower actuation. While $\lambda_0$ is influenced by the interaction of the tether force and the gravitational and aerodynamic forces acting on the individual kite components, $\alpha_\mathrm{d}$ is inversely related to the relative power setting $u_\mathrm{p}$ and correlated with a spanwise twisting and bending of the bridled membrane wing.

The measurements show that the lift-to-drag ratio of the kite increases with the relative power setting. For straight flight the maximum ratio is reached at an angle of attack of $8°$ and a moderate lift coefficient. Steering maneuvers reduce the lift-to-drag ratio. For the investigated data set the variation of the angle of attack during the traction phases is limited to about $8°$. Because of the constant force control operation of the ground station, the angle of attack is inversely related to the relative flow velocity. During the traction phase, the angle of attack and the lift coefficient are both increased, yet strongly influenced by the effect of gravity, which varies with the heading of the kite. When flying upwards, the flight speed of the kite decreases and the angle of attack increases to compensate for the effect of gravity, when flying downwards, the speed increases and the angle decreases.

We find that the aerodynamic characteristics of the bridled membrane wing do not only depend on the angle of attack, as common for rigid wings, but also on the level of aerodynamic loading. For the investigated C-shaped wing, we observe, for example, that increasing the loading causes the wing to flatten which enlarges the projected area and amplifies the effective aerodynamic force. Because the loading is actively controlled by the relative power setting, we can use this parameter to correlate the effect of the loading on the aerodynamic characteristics. How exactly the power setting affects the wing shape depends strongly on the layout of the bridle line system. Our measurements show that accounting only for the dependency on the angle of attack variation, as commonly done, leads to a considerable uncertainty of the aerodynamic characteristics. We

expect that using the relative power setting as a secondary influencing parameter will improve the aerodynamic characterization of a pumping cycle AWES over the whole flight envelope.

Using a Kalman filter, as shown by Schmidt et al. (2017), could help to increase the accuracy of the measured data and the actual state of the system by including knowledge about system behavior and the data of other onboard sensors, such as the inertial measurement unit. Another possible improvement of the analysis would be to retrofit the power and steering lines with force sensors, to assess the aerodynamic load distribution on the wing. The effect of the kite control unit is considerable as it contributes about 40% of the total kite mass and 10% of the drag. Because of the suspension in the steering lines it can exhibit unpredictable movements, particularly during turns and when the tether force is low during retraction phase. This adds uncertainty to the calculated orientation of the kite. Moving the control unit towards the bridle point and connecting it to power and steering lines could potentially avoid this problem.

*Data availability.* The data set used for this study is available from https://github.com/rschmehl/wes2018. Data sets of current test flights can be requested freely for research purposes by emailing Kitepower B.V. at data@kitepower.nl.

## Appendix A: Flow velocity induced by the wing at the measurement location

To estimate the aerodynamic effect on the relative flow at the measurement location below the wing we introduce a lifting line approximation of the C-shaped wing, as depicted in Fig. A1. The center section is represented as a straight vortex segment

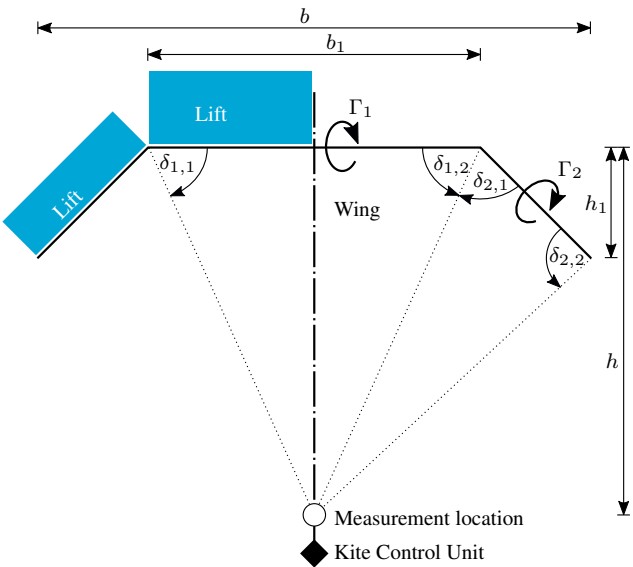

**Figure A1.** Front view of the lifting line model of the kite, flying towards the reader.

of finite length with circulation $\Gamma_1$, while the tip sections are represented by straight segments with circulation $\Gamma_2$. The con-

tribution of a vortex segment $i$ to the induced velocity $v_{\mathrm{ind}}$ at the measurement location is calculated by the Biot-Savart law

$$v_{\mathrm{ind},i} = \frac{\Gamma_i}{4\pi r_i}\left(\cos\delta_{i,1} + \cos\delta_{i,2}\right), \tag{A1}$$

where $r_i$ is the perpendicular distance between the measurement location and the vortex segment, $\delta_{i,1}$ and $\delta_{i,2}$ are the angles
between the segment and the (dotted) lines connecting its ends with the measurement location. Following the superposition
law of potential flow, the velocity induced by the three vortex segments at the measurement location is calculated as

$$v_{\mathrm{ind}} = v_{\mathrm{ind},1} + 2v_{\mathrm{ind},2}. \tag{A2}$$

Because of the symmetric layout of the wing with respect to the measurement location, we first calculate the velocity induced
by one wing half and then double this effect. Using the geometric parameters defined in Fig. A1 we can determine the line
angles $\delta$ and from these calculate the velocity induced by the center section

$$v_{\mathrm{ind},1} = 2\frac{\Gamma_1}{4\pi h}\cos\left[\arctan\left(\frac{2h}{b_1}\right)\right], \tag{A3}$$

and by each tip section

$$v_{\mathrm{ind},2} = \frac{\sqrt{2}\Gamma_2}{4\pi\left(h+\frac{b_1}{2}\right)}\left\{\cos\left[\frac{3}{4}\pi - \arctan\left(\frac{2h}{b_1}\right)\right] - \cos\left[\frac{3}{4}\pi - \arctan\left(\frac{2(h-h_1)}{b}\right)\right]\right\}. \tag{A4}$$

The apparent flow velocity $v_{\mathrm{a}}$ at the measurement location is the sum of the freestream velocity $v_\infty$ and the induced velocity
$v_{\mathrm{ind}}$. Using the values listed in Table A1 we calculate an induced velocity $v_{\mathrm{ind}} < 0.2$ m/s. Considering the induced downwash

**Table A1.** Model parameters of the LEI V3 kite for lifting-line theory

| $b$ | $b_1$ | $h$ | $h_1$ | $\Gamma_2/\Gamma_1$ | $L$ | $\overline{v_\infty}$ |
|-----|-------|-----|-------|---------------------|-----|-----------------------|
| 8 m | 4 m | 8.5 m | 2 m | 5/8 | 3250 N | 18 m/s |

due to the wing tip vortices, the induced angle of attack at the measurement location is $\alpha_{\mathrm{ind}} < 0.6°$. Accordingly, the effect
of the induced velocity on the relative flow measurements is negligible and the apparent flow velocity can be considered to
be equal to the free stream velocity. For larger kites with much higher tether forces the induced velocity might play a more
important role and should thus be taken into account for precise measurement of the relative flow.

*Competing interests.* Roland Schmehl is co-founder of and advisor for the start-up company Kitepower B.V. which commercially develops a
100 kW kite power system and which provided their test facilities and staff for performing the in situ measurements described in this article.
Both authors are financially supported by the European Union's Horizon 2020 project REACH, which also provides funding for Kitepower
B.V.

*Acknowledgements.* We would like to express our gratitude to the colleagues from Kitepower B.V. who provided their expertise and hardware which enabled us to conduct this research. This project has received funding from the European Union's Horizon 2020 research and innovation programme under the Marie Skłodowska-Curie grant agreement No. 642682 for the ITN project AWESCO and the grant agreement No. 691173 for the "Fast Track to Innovation" project REACH.

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
