# Peer review of "Aerodynamic characterization of a soft kite by in situ flow measurement"

_Wind Energy Science, 2018_

## Short Comment (SC1) · 28 Jul 2018

This manuscript tackles a hot and important topic for the airborne wind energy community: the aerodynamic identification of soft kites. As firmly explained in the Introduction, it is a hard problem due to the intrinsic difficulties arising in both CFD and experimental analysis. In this regards, this manuscript is helpful because it highlights the importance of aeroelastic effects and their consequences on the aerodynamic coefficients for later use in AWE dynamic simulators and optimizers. I fully agree with the main conclusion of this work in the sense that a good aerodynamic characterization of a soft kite should take into account the deformation of the kite.

Regarding the accuracy of the experimental data, which exhibits certain degree of dispersion, is difficult for me to make an assessment because the methodology is far from my expertise at UC3M. The authors used vanes attached to the two power lines and a quasi-steady model that relies on several assumptions to estimate the forces. Our group is working on a multi-hole probe on-board a long pole attached to the central strut and an extended Kalman filter to estimate the aerodynamic forces and torques. I agree with the authors that the two power lines are generally well-tensioned but I am not sure that the plane spanned by them is a good choice for characterizing the orientation of the kite. In any case, it is necessary to try different methods, compare them, and learn about their advantages and disadvantages. From this perspective, it would be interesting to mount both experimental setups in the same kite and compare the measurements of all the instruments.

Minor comment 1: $\beta$ is normally used to denote the side-slip angle. Several works on AWE systems used $\Gamma$ for the elevation angle of the tether.

Minor comment 2: if possible, a diagram $C_L$ versus $C_D$ would be interesting. How far is the drag coefficient from a polar $C_D = C_{D0} + kC_L^2$ ?

———————————————

---

## Short Comment (SC2) · 30 Jul 2018

[revised manuscript text omitted]

---

## Short Comment (SC3) · 2 Aug 2018

[revised manuscript text omitted]

---

## Short Comment (SC4) · 6 Aug 2018

Thank you for your comment. This answer will deal with the main question 'mounting of the sensors' and the two minor comments seperately.

Firstly, mounting of the sensors: The rigid frame attached to the front bridle lines, also called power lines remained our favourite choice after exploring several other methods. In Fig. 1 of the paper you see a previous attempt with additional lines and a drag shuttle which proved to require much time for installation and did not yield satisfying results throughout the flight. We also used the same sensors described in this publication (wind vanes and Pitot tube) in a more conventional, longitudinal air data boom setting

in the symmetry plane of a surf kite. The setup is described in our recent publication [Oehler, J., van Reijen, M., and Schmehl, R.: Experimental investigation of soft kite performance during turning maneuvers, Journal of Physics: Conference Series, 1037, 052 004, http://stacks.iop.org/1742-6596/1037/i=5/a=052004, 2018.], see also Fig. 1 attached to this comment.

Such a mounting has the advantage that the sensor setup is always aligned with the center chord of the kite which avoids the calculation/estimation of a 'depower angle' $\alpha_d$. However the center chord and also the possibilities to attach an air data boom vary from kite to kite. Some do have an inflated strut at the center, some (like our kite in Fig. 1 of this comment) have struts left and right to the center chord. As consequence such a mounting is more dependent on the particular kite model than the mounting in the power lines which we chose. Another clear advantage of our current mounting is that we could directly derive the kite's lift-to-drag ratio. The suggested comparision of two or more different mountings during one flight which you suggest would of course be ideal to derive a better conclusion on the pros and cons of different sensor mounging strategies, however this was not undertaken in this study.

Secondly, elevation angle: We chose $\beta$ mainly to be consistent with the publications of our own group where $\beta$ is used for elevation and $\beta_s$ for the sideslip angle. We also found $\theta$ widely used as polar angle but since there was no dominant notation we chose to keep it with this notation. $\Gamma$ is in aerodynamic literature common for the circulation which we also used for this paper.

Thirdly, polar for lift and drag coefficient: We did not include such a plot but it would for sure be worth a further study. We did not measure lift and drag separately but only the lift-to-drag ratio directly. Since the lift-to-drag ratio (see Fig. 15) shows a behaviour which is qualitatively similar to the one of a profile or 3D airplane we would expect to get a curve which is close to a parabolic curve. Lift-to-drag ratio has its maximum at a medium angle of attack and lift coefficient and is lower for low angles of attack and high angles of attack. This is similar to what we expect for a 2D profile.

What hinders us from measuring a full polar is that we cannot freely chose angle of attack and lift coefficient like it can be done for profiles or aircraft models in a wind tunnel or full aircraft with the elevator. A kite only flies and keeps its shape as long as it creates a considerable amount of lift. Measuring drag and lift at zero lift condition or for very low $C_L$ is therefore not feasible. Specifically for our test setup and the flight described, the airspeed measurement was of low quality for low flight speeds. Since all the flights in depowered state with low lift, low drag and low lift-to-drag ratio show low flight speeds we could not measure this part of a lift drag polar. Since lift and drag coefficients depend crucially on precise measurement of the flow velocity we did not calculate and plot these values for the depowered kite. In Fig. 16 and 17 the lift coefficient $C_L$ is only shown for the powered kite with comparably high angle of attack and lift coefficient.

Since different angles of attack are usually achieved by changing the power setting which deforms the kite and further higher loading changes the shape of a kite, the polar would need an additional parameter describing the deformation for a complete description. Similar parametrized polars are used for rigid polars which have e.g. Reynolds number or flap extension included. This requires first a better understanding or at least a common parametrization of the deformation of a soft kite.

[Figure]

**Fig. 1.** Center chord parallel air data boom as it was used in [Oehler, J., van Reijen, M., and Schmehl, R.: Experimental investigation of soft kite performance during turning maneuvers]

---

## Author Comment (AC1) · 6 Aug 2018

We have provided video footage to supplement Figure 4, showing still images from the mast-based launch,

https://www.youtube.com/watch?v=uexBpa7ovts
https://www.youtube.com/watch?v=l2es08DBu6s
https://www.youtube.com/watch?v=noZ38Vj0Ld8
https://www.youtube.com/watch?v=rY1yvzaQYH4
https://www.youtube.com/watch?v=gMpwjRqe0g4
https://www.youtube.com/watch?v=k50HvzitpzQ

[Figure]

We have further provided video footage to supplement Figure 7, showing still images of the wing, filmed from the kite control unit,

https://www.youtube.com/watch?v=KoPYF3UDXQ8

and still images of the wing filmed from the other wing tip,

https://www.youtube.com/watch?v=TsL3Qy7Q1i0

a figure eight maneuver filmed from the ground,

https://www.youtube.com/watch?v=IaNjyol_qtQ

and a similar maneuver filmed from a camera drone

https://www.youtube.com/watch?v=3wwaE7Ul2DY
* * *

---

## Short Comment (SC5) · 15 Aug 2018

This paper addresses an important topic for the further development of AWE technology: the aerodynamic characterization of soft kites, such as the leading-edge inflated (LEI) tube and the ram-air kites. Even though a significant number of players in the AWE community have migrated in recent years to "rigid" (airplane-like) wings, AWE concepts based on soft kites continue to be developed and studied by some groups due to advantages such as low materials cost and facilitated transportation, assembly and maintenance/replacement.

As well pointed out by the authors, differently from what is common for the rigid wing

case, wind tunnel tests are very difficult (if not unfeasible) to be carried out for the identification of the coefficients of aerodynamic lift and drag of a soft kite. However, for tasks such as flight trajectory optimization, it is of fundamental importance to know such aerodynamic properties of the kite. In other words, one question to be answered is: "how do the angle of attack and the coefficients of lift and drag (which impact the tether traction force and hence the reel-out/in power) depend on control inputs such as the steering and the powering/pitching commands"? Understanding how these variables are correlated would allow us to design more efficient and robust flight trajectories in a pumping cycle.

This paper, which is rich in technical details about the prototype setup, methodology and discussion of the results, sheds some light on the problem based on field-test data with LEI tube kites. Plots of the lift-to-drag ratio (CL/CD) and the lift coefficient (CL) are presented and discussed, in real flight situations involving different heading angles, apparent wind speeds, powering settings and steering inputs. The presented results seem to corroborate a behavior so far assumed for the soft kites: as the powering setting is reduced in order for the reel-in phase to begin, the CL/CD falls, allowing for the tether to be recovered with only a minor expense of energy. Also, the behavior of the CL/CD ratio as function of the angle of attack seems to roughly match, in a span from -5 deg to 15 deg, some curves found in the literature and so far assumed representative for soft wings, which were adapted from the rigid wing case in an ad-hoc fashion. Also, from the results it can be clearly seen that steering commands of high intensity cause a significant decrease in CL/CD as well, which is something to keep in mind when defining the curvatures of the flight trajectories in the reel-out phase (usually a "lying eight" figure).

In my opinion the analysis performed in this work could be further refined by using filters, such as the Kalman Filter and its variations (the Extended and the Unscented Kalman Filter, for instance). The idea is that these filters incorporate what is known about the system (flight) dynamics, perhaps based on dynamic models such as the

point-mass kite. In this way not only problems such as outliers in the plots could be better handled, but also the uncertainty/variance in the variables directly measured could be minimized. The more inputs (measurements) are available (fed) to the filter, the better is the filtering performance. A preliminary work in this direction was carried out by Schmidt et. al. (2017).

Congratulations to the authors on the excellent work.

References:

E. Schmidt, M. De Lellis, R. Saraiva, and A. Trofino. State estimation of a tethered airfoil for monitoring, control and optimization. In Proceedings of the 20th IFAC World Congress, volume 50–1, pages 13246 – 13251, Toulouse, France, July 2017. IFAC. doi: 10.1016/j.ifacol.2017.08.1960

---

## Short Comment (SC6) · 20 Aug 2018

Thank you for your interest in our work and also for your feedback and assessment of our findings. As you wrote in your comment our focus was the data acquisition and to find a good way for measuring the apparent flow variables. Implementing a Kalman filter is without doubt a very good idea for a follow up study but was out of scope for this work. The kite research group in Madrid (UC3M) with Ricardo Borobia-Moreno and Gonzalo Sanchez-Arriaga is also working on this topic so it seems appropriate to include your input in the conclusion section of our publication.
You write in your cited article that you could extend your current filter to accommodate

the use of airborne data in addition to ground based measurements. In case you are interested in using our data set which includes position, attitude and accelerations of the kite plus the aerodynamic measurements you are kindly invited to do so.

---

## Referee Comment (RC1) · Anonymous Referee #1 · 22 Aug 2018

The paper describes an experimental approach to estimate some basic aerodynamic and performance characteristics of a soft kite that is used for airborne wind energy generation. This is achieved by applying a novel setup for measuring airspeed, angle of attack and angle of sideslip in a position between the power lines of the kite in a short distance above the kite control unit. For the presented soft kite this setup seems to fulfill the premises to obtain meaningful information about the relative airflow at the kite. From these measurements, the measured tether force and elevation angle, together with systems parameters from geometry and masses, the authors were able to estimate L/D, CL and the angle of attack at the chord. Thereby the could show that the variations of angle of attack and the angle of sideslip are not as large as indicated in

the literature. They also could approximately reproduce the magnitude of L/D derived from aerodynamic models.

Although I am not very familiar with the literature and recent achievements in the AWE domain, it seems to me that their approach using the AWESOME measurement equipment offers new possibilities in obtaining valuable data for the characterisation and modeling of AWE soft kite systems. The possibility of measuring the angle of sideslip for such a system is unique. I see a high potential for future use. The content of the paper is good and worth to be archived. Nevertheless there are a few aspects I would like to comment on.

Specific comments:

As the authors discuss many simplifications they seem to be aware of the limitations of their results. Many of their assumptions are subject of significant uncertainties. To name a few: They use a fixed geometry derived from a CAD model, although due to the flexibility and elasticity of the system, the assumed geometry of bridles, lines and canopy is deformed depending on the changing loads acting on these elements. Another significant simplification is the assumption of flying in quasi-steady equilibrium. From what I know, crosswind-trajectories are highly dynamical maneouvres and accordingly not only the gravitational but also the inertial forces and moments have to be taken into account. They also consider unsteady airflow when discussing the oscillations observed. The assumption of a fixed center of pressure is a massive simplification too. On the other hand, it is comprehensible to simplify, because such effects are much more difficult to account for. Nevertheless, in my opinion, simplifications and neglected effects should be especially used in the discussion of the results, for example to explain the large dispersion of the derived L/D and CL. Obviously (see fig 15 and 17) the applied filters alone were not able to reduce the dispersion very much. In my opinion the discussion and explanation can be improved here.

Concerning the results (fig 15) I did not understand the trend of L/D vs alpha for the depowered flight. As noted, it contradicts the trend in the aerodynamic models. Although it is said that the angle of attack doesn't have strong effect in this flight regime as the wing is largely deformed, it does not explain the clear trend of L/D being reduced with increasing alpha. If possible, an explanation for the opposite trend should be provided.

Further comments and technical corrections:

1) Right at the beginning it is said that wind tunnel testing of large deformable kites is practically not feasible. It is possible, but of course it is a question of money and available facilities. In US large gliding parachutes have been tested in the wind tunnel (see Geiger/Wailes: Advanced Recovery System Wind Tunnel test Report, NASA TM CR 177563, 1990). In Europe a scaled model of the FASTWing parachute was tested in the DNW-LLF wind tunnel (see Willemsen et al: The FASTWing project: Wind Tunnel Tests, Realization and Results, AIAA 2005-1641).

2) On page 3 the power setting "up" is introduced but not clearly defined. An implicit definition is later provided in equation 5. On page 3 a reference to eq. 5 should be included.

3) On page 9 "cref" is defined perpendicular to the power line, but in fig. 5 "cref" seems to be defined as horizontal distance. Please update fig. 5.

4) On page 10 the authors refer to a "mechanistic model". Does this mean a rigid body model? Please explain the meaning.

5) The calculation of "lambda0" is discussed on pages 12 and 13. Here the corresponding equations should be given.

6) "Beta" is usually used for the angle of sideslip. If possible use a different symbol for the elevation angle.

Again, the paper is good. It only needs a minor revision.

---

## Short Comment (SC7) · 24 Aug 2018

I enjoyed reading the paper and I agree with other comments that the paper has interesting contributions and deserves to be published. I also think that the use of dynamic estimators, like Extended Kalman Filter, could help to improve the characterisation of the kite aerodynamic coefficients and the proposed measurements for the apparent flow variables will certainly improve the filter estimates. The possibility of using your data set from my research group, as you suggested, is very interesting and could be a first step towards a future collaboration.

---

## Referee Comment (RC2) · Anonymous Referee #2 · 27 Aug 2018

This is an interesting paper that presents a novel method of measuring the lift-to-drag of kites used for wind energy extraction. The experimental setup consists of a rigid frame with mounted wind velocity and direction sensors. The frame is attached to the power lines of the kite. The reported measurements include angle of attack, side slip angle, and lift-to-drag, and they appear to be reasonable for in-situ measurements. The main concern of this referee is the stated strong fluctuations at a frequency of 1.2 Hz, corresponding to a reduced frequency of k=0.5. This indicates a highly unsteady flow pattern with substantial hysteresis in and lift and drag curves. To address the observed fluctuations, authors smooth the data by averaging over 3 cycles. It is unclear why the data were not phase averaged? Phase averaging would allow a more sensible approach for processing the data, and furthermore, the measured aerodynamic values can be compared with those for solid airfoils at the same reduced frequency. Publication would be merited once the authors respond to this key issue.

---

## Short Comment (SC8) · 29 Aug 2018

In the paper a technique to measure the flow direction and speed for a soft kite is introduced. The flow sensors are attached below the kite in the bridle lines. To compensate for bridle sagging a simple force balance is solved. The measured angles and flow velocity show oscillatory behaviour which originates from the ground station. The oscillations induce highly unsteady aerodynamic behaviour and, in order to reduce this effect the measured data is smoothed over three periods. The resulting glide ratio during flight is presented and exhibits a strong dependence on power setting, and heading angle. Overall this paper shows a clear methodology and interesting results from a

flight test. In the reviewer's opinion it is ready for publication with minor changes.

Page 13, lines 6-8: "The effect is different for every kite/bridle combination but a ratio of 3:1 for the forces in front and back bridles seems like a good average value."

Is there a source to cite or an argument to support this statement?

Page 15: "In order to minimize this effect, the data is smoothed over an interval of T = 2.5s which is equivalent to 3 periods of the oscillation."

From a practical point of view the smoothing is a simple and logical approach to obtain results. Can the authors comment on the magnitude of error introduced by the smoothing, e.g. compared to other filtering techniques?

Page 22: "A change in power setting causes a complex deformation of the wing and thereby affects the aerodynamic coefficients, while a change in angle of attack affects the aerodynamic coefficients by changing the flow field."

This statement is not entirely true. As stated in the introduction the soft kite also deforms during flight for different flight speed and angles of attack. Nevertheless, it is correct to assume two different wings for powered and de-powered configuration.

---

## Author Comment (AC2) · 31 Aug 2018

Thank you for reading the submitted manuscript and for your comments and recommendations. We will take these into account when compiling the revised version of the manuscript. And, of course, we are also looking forward to a collaboration with your research group.

---

## Author Comment (AC3) · 1 Sep 2018

Thank you for your interest in our work and your critical review.

Your first question on the force fraction of force transmitted via back bridle lines to front bridle lines is a very justified one since the pressure point of kites generally does not stay in one position for different power settings. Since there was no data available for the kite model we used and our setup with KCU and kite flown on a single main tether was not equipped to measure it we could not track the actual force fraction during the flight. The attached graph from Jan Hummel's PhD thesis 'Automatic measurement and characterization of the dynamic properties of tethered flexible wings' (TU Berlin 2017)

[Figure]

shows the force fraction $f$ over $\epsilon_{rel}$ ('relativer Powerweg' is equivalent to $u_p$). Although we can see a trend to higher values of $f$ for higher power settings $u_p$ we decided for a constant value of $f = 0.33$ because the relation or magnitude of this trend for our kite remains unknown. Further was the relation only measured for a four line kite in static flight whereas we used a kite with a single main tether and flew crosswind maneuvers.

Regarding the smoothing, we did not try different methods so we cannot compare to other filtering techniques. Initially we only used a smoothing period of $0.3s$ to eliminate noise of the sensors but this left us with the problem that the mentioned high frequent oscillations of the kite where inertia plays an important role would render the assumption of a force equilibrium inapplicable.

With your third comment you are of course right. Due to higher loading or different flow conditions also flight speed and angle of attack do have an effect on the shape of the kite. We write 'a change in angle of attack affects the aerodynamic coefficients by changing the flow field' but we do not claim that it does not affect the shape of the kite. In case of a change in angle of attack we primarily change the flow field which affects the shape of the kite whereas if we change the power setting it is in terms of logic 'the other way round'. But of course we have to always consider that both flow field and shape of the kite are strongly coupled and we can never change one without affecting the other.
* * *
[Figure]

**Fig. 1.** Force fraction between back lines and front lines over power setting for different kites ['Automatic measurement and characterization of the dynamic properties of tethered flexible wings' ,Jan Hummel]

[Figure]

---

## Author Comment (AC4) · 3 Sep 2018

The main reasons for not using phase averaging were that both authors are not familiar with this method and that the observed fluctuations were not given a very high priority since they were seen rather as misbehavior of the system. The oscillations at $f = 1.2$ Hz did not occur during the whole flight but only at several occasions for intervals of about 10 seconds. The amplitude always stayed below 2 degrees for the angle of attack and 1 m/s for the flow velocity. Figure 10 shows the most pronounced oscillation of this kind. Since those fluctuations are caused by a bad response of the ground station and do not occur in a systematic way we did not focus on this phenomenon. The variations

that both flight speed and angle of attack undergo while the kite performs its pumping cycles show a much larger amplitude. Since we did not have experience in applying phase averaging and the oscillation is more regarded as an unwanted anomaly we decided for a simple "moving average" filter.

When looking at the repeated scheme of controlled figures of eight or entire pumping cycles there might be a different picture. One cycle takes about 2-3 minutes and one figure of eight about 40 seconds. If phase averaging could help to fuse the data of different cycles or different figures of eight into useful information and increase statistical relevance it would be a great tool. In our case we had a quite limited number of only 10 different pumping cycles which also showed large variations in flight path, power setting and wind speeds.

A more detailed explanation of the oscillation modes of the kite can be found in http://awec2017.com/images/posters/Poster_Oehler.pdf which is also referred to in the paper.

---

## Author Comment (AC5) · 3 Sep 2018

We respond to the referee comment by including our answers in a copy of the original comment:

The paper describes an experimental approach to estimate some basic aerodynamic and performance characteristics of a soft kite that is used for airborne wind energy generation. This is achieved by applying a novel setup for measuring airspeed, angle of attack and angle of sideslip in a position between the power lines of the kite in a short distance above the kite control unit. For the presented soft kite this setup seems to fulfill the premises to obtain meaningful information about the relative airflow

at the kite. From these measurements, the measured tether force and elevation angle, together with systems parameters from geometry and masses, the authors were able to estimate L/D, CL and the angle of attack at the chord. Thereby the could show that the variations of angle of attack and the angle of sideslip are not as large as indicated in the literature. They also could approximately reproduce the magnitude of L/D derived from aerodynamic models.

Although I am not very familiar with the literature and recent achievements in the AWE domain, it seems to me that their approach using the AWESOME measurement equipment offers new possibilities in obtaining valuable data for the characterization and modeling of AWE soft kite systems. The possibility of measuring the angle of sideslip for such a system is unique. I see a high potential for future use. The content of the paper is good and worth to be archived. Nevertheless there are a few aspects I would like to comment on.

**Specific comments:**

As the authors discuss many simplifications they seem to be aware of the limitations of their results. Many of their assumptions are subject of significant uncertainties. To name a few: They use a fixed geometry derived from a CAD model, although due to the flexibility and elasticity of the system, the assumed geometry of bridles, lines and canopy is deformed depending on the changing loads acting on these elements. Another significant simplification is the assumption of flying in quasi-steady equilibrium. From what I know, crosswind trajectories are highly dynamical maneouvres and accordingly not only the gravitational but also the inertial forces and moments have to be taken into account. They also consider unsteady airflow when discussing the oscillations observed. The assumption of a fixed center of pressure is a massive simplification too. On the other hand, it is comprehensible to simplify, because such effects are much more difficult to account for. Nevertheless, in my opinion, simplifications and neglected effects should be especially used in the discussion of the results, for example to explain the large dispersion of the derived L/D and CL. Obviously (see

fig 15 and 17) the applied filters alone were not able to reduce the dispersion very much. In my opinion the discussion and explanation can be improved here.

**Response:** Thank you for this input. Indeed the simplifications that we made in our modeling assumptions should be given more space in the discussion of results. We will consider this in the final publication.

Concerning the results (fig 15) I did not understand the trend of L/D vs alpha for the depowered flight. As noted, it contradicts the trend in the aerodynamic models. Although it is said that the angle of attack does not have strong effect in this flight regime as the wing is largely deformed, it does not explain the clear trend of L/D being reduced with increasing alpha. If possible, an explanation for the opposite trend should be provided.

**Response:** The mentioned effect that L/D is reduced for a higher angle of attack is only visible when looking at one certain power setting (e.g. the lowest one in dark blue in fig. 12). Since the measured inflow angle itself is used for the calculation of L/D a change in the measured flow angle causes this change in L/D. A steering command or change in the heading of the kite can be the root cause of this. L/D is actually not reduced with increasing alpha for depowered flight. This can best be seen in fig. 12. The main trend is a higher L/D value for a higher power setting at higher angles of attack.

**Further comments and technical corrections:**

1) Right at the beginning it is said that wind tunnel testing of large deformable kites is practically not feasible. It is possible, but of course it is a question of money and available facilities. In US large gliding parachutes have been tested in the wind tunnel (see Geiger/Wailes: Advanced Recovery System Wind Tunnel test Report, NASA TM CR 177563, 1990). In Europe a scaled model of the FASTWing parachute was tested in the DNW-LLF wind tunnel (see Willemsen et al: The FASTWing project: Wind Tunnel

Tests, Realization and Results, AIAA 2005-1641).
**Response:** We agree and have adjusted the manuscript accordingly, a.o. by referring to the experimental campaigns at NASA Ames and DNW-LLF.

2) On page 3 the power setting "up" is introduced but not clearly defined. An implicit definition is later provided in equation 5. On page 3 a reference to eq. 5 should be included.
**Response:** We write after the introduction of $u_p$ "A high value of $u_p$ means that the wing is powered, while a low value of up means that the wing is depowered.". We agree that refering to Eq. 5 where $u_p$ is linked to the geometric lengthening of the rear bridles is better. We will include this reference.

3) On page 9 "cref" is defined perpendicular to the power line, but in fig. 5 "cref" seems to be defined as horizontal distance. Please update fig. 5.
**Response:** The reference chord length $c_{ref}$ is a geometric parameter of the wing which we use to relate the actuation of the steering lines $\Delta l$ and the pitch rotation of the wing, quantified by the depower angle $\alpha_d$. We have slightly reworked the definition of $c_{ref}$. It measures the distance from the trailing edge of the wing, where the steering lines are attached, to the (virtual) point of rotation of the wing. Please check the revised manuscript for more details. The state displayed in Fig. 5 is a rotated (depowered) state with $\alpha_d > 0$. We intentionally chose a perspective at which the reference chord is horizontal relative to the paper.

4) On page 10 the authors refer to a "mechanistic model". Does this mean a rigid body model? Please explain the meaning.
**Response:** With "mechanistic model" we actually mean the geometric model discussed in Fig. 5. We have adjusted this in the text.

5) The calculation of "lambda0" is discussed on pages 12 and 13. Here the corresponding equations should be given.
**Response:** Since we use a numerical solver for the bridle line angles the whole algorithm would be rather long. We updated the text already to say that we use a numerical shooting method upon the request of one reviewer. We added a description of the algorithm.

6) "Beta" is usually used for the angle of sideslip. If possible use a different symbol for the elevation angle.
**Response:** We chose $\beta$ mainly to be consistent with the publications of our own group where $\beta$ is used for elevation and $\beta_s$ for the sideslip angle.

---

## Author Comment (AC6) · 11 Sep 2018

There are additional reasons why we did not use phase averaging in this study. In general, harvesting wind energy with tethered flying devices has more degrees of freedom than harvesting with towered wind turbines. That is, because the tether provides only a maximum distance constraint in the radial direction, between ground station and wing. During a pumping cycle, the wing is actively steered and powered/depowered while the ground station actively controls the reeling speed of the tether. Because of its short response time, the ground station is generally used to control the maximum tether force, which is of particular importance when flying fast, with a relatively lightweight wing

trough a fluctuating wind environment.

On the other hand, the ability to adjust the harvesting altitude and flight maneuvers to the instantaneous wind resource is also one of the major advantages of airborne wind energy. Because of the strong dependency of the flight operation on the variable wind environment, the flight trajectories generally vary significantly. This was the case for the dataset that we acquired for the present study. The flight trajectories of the different pumping cycles differed to a degree that it was challenging to consistently determine a phase location. For a wind turbine, with rotor blades that are mechanically linked and comparatively large rotational inertia, this is much more straightforward. For this reason, we decided to present the data in the way that it is now implemented in the manuscript. At this stage of development of the measurement setup, a rigorous phase averaging would have introduced more uncertainties, in our opinion. What we did instead is to distinguish between the reel-in and reel-out phases, subdividing the crosswind maneuvers further into flying up (against gravity) and flying down down (with gravity). In our view this can already be regarded as a first step in the direction of phase averaging, but customized based on the specific physics of tethered flight in pumping cycles.

For practical reasons, our dataset covered only 5 separate pumping cycles, which was by far not sufficient to perform any meaningful statistical analysis. This was another reason for us to avoid a more thorough statistical analysis.

We will incorporate the above reasoning in the revised manuscript.

---

## Short Comment (SC9) · 13 Sep 2018

This paper deals with aerodynamic characterisation of a soft kite. A lot of work has been done until now on soft kites, especially on kite control along a given trajectory and on the experimental measurements of kites. The data obtained from measurements are then very useful to compare and adjust kite modelling. For example, several models of kite flight like "zero-mass model" or "point-mass model" were developed and integrated in kite control systems. All these models need the aerodynamic coefficients (lift, drag, fineness) as inputs. The most sophisticated models may use lift and drag polars to take into account the variation of the lift and drag coefficients with angle of

attack. Unfortunately, it is very difficult to obtain good estimations of the lift and drag coefficients. The measurements were generally made in static flight and then integrated to dynamic models. Nevertheless, there are big differences between lift and drag coefficient in static and dynamic flight conditions. Therefore, several numerical models were developed in order to obtain the aerodynamic characteristics of a kite. The most sophisticated models are based on Fluid Structure Interaction taking into account as much as possible the real geometry of the kite. For example, the model developed by Maison et al. (A. MAISON, A. NÊME, J.-B. LEROUX, De la problématique du dimensionnement de grands kites, ATMA, 2017) for example takes into account the aeroelastic behaviour of the canopy, with an orthotropic modelling of the fabric but also the inflatable structure, the bridles and the lines. Unfortunately, these models are very difficult to validate due to the lake of measured data. Therefore, the work presented in this paper is very interesting for the validation of kite numerical models. In this work, a relatively complete set of measures were performed. The apparent wind velocity is measured using pitot tubes and the attack and drift angles were measured. This paper points out the fact that the angle of attack may substantially vary during flight manoeuvres. One fact that is shown in this paper is that the lift to drag ratio drops during turning phases. This phenomenon that was expected is already studied at ENSTA Bretagne, and these data will be very useful for the validation of these models. Another interesting point is that the angle of attack will adjust depending on the orientation of the kite within the wind window. All the measured data show the importance of taking into account the aerodynamic characteristics of a kite more precisely in the kite flight modellings. Therefore, these data will be very helpful to improve the kite flight models and the control laws. These data will also be useful for the validation of Fluid Structure Modelling of Kites.

Remarks p13 Back line force Measurement: It may be possible to calculate the backline force from measurement of instantaneous power consumption of the motors and knowing the rotation velocity of actuator. Is this method feasible? This would allow a more precise estimation of the shift in center of pressure. p13 Was the ratio of 3:1

for the forces in front and back bridles already measured on the kite? For example, in (Xaver Paulig, Merlin Bungart, Bernd Specht, "Conceptual Design of Textile Kites Considering Overall System Performance", AWE Book, 2013), a ratio of 10:1 was measured on a parafoil kite. Recently, a ratio of 2:1-1:1 was measured by Behrel in ("Investigation of kites for auxiliary ship propulsion: experimental set-up, trials, data analysis and kite specs novel identification approach", Ph.D. Thesis, 2017) p13, l.10 senors' => sensors p13 Is it possible to have the formula to calculate lambda1 and lambda2? p14 How is determined the projected surface because it may vary during kite flight?

General Remarks: Do you have access to Kite modelling data, and more precisely on the lift and drag coefficient obtained? Can you compare it with the measured data?

Is it feasible in the future to measure the wind velocity with a 3 directions ultrasonic sensor in order to have more precise data?

---

## Author Comment (AC7) · 17 Sep 2018

1) Remarks p13 Back line force Measurement.

We have not measured the force ratio within this study about in situ aerodynamic measurement of a kite used for pumping cycle operation in an AWES. We have, however, measured this ratio in a separate study about a similar but smaller kite flown by a human pilot on the beach. The corresponding MSc thesis is reference and can be accessed here: van Reijen, M.: The turning of kites, Master's thesis, Delft University of Technology, http://resolver.tudelft.nl/uuid:5836c754-68d3-477a-be32-8e1878f85eac, 2018. The measured force ratio is plot-

ted in Figure 6.8 and ranges from 0.75 down to 0.45 (see Fig. 1 in this response). This seems to correspond well to the finding of Behrel (2017), who uses the same kite design (LEI tube kite). We will include more of this information in the revised manuscript. $\lambda_1$ is an estimated value, while $\lambda_2$ is a solution of the force equilibrium. We did not include the relatively simple and straightforward algorithm to not expand the already extensive manuscript much further. We are not explicitly accounting for the variation of the projected surface of the kite but instead use the constant value of the CAD geometry as reference value.

2) General Remarks: Do you have access to Kite modelling data, and more precisely on the lift and drag coefficient obtained? Can you compare it with the measured data?

The objective of the present study was to experimentally determine the aerodynamic characteristics of the kite, during flight operation in pumping cycles. In our opinion, such aerodynamic data of high quality is one of the most critical input data for kite models. Including also a model in the present study, and describing this to the required level of detail, would have been out of scope of this already quite extensive study.

3) Is it feasible in the future to measure the wind velocity with a 3 directions ultrasonic sensor in order to have more precise data? We have assessed this technology but concluded that presently, the simple Pitot tube with two orthogonal flow vanes is the most robust setup. Next to the sensors themselves, the key challenge is to find a relatively stable reference frame, because the KCU is quite dynamic and the wing is deforming. In our opinion, the plane spanned by the two power lines is the best location to attach a sensor.

**Fig. 1.** Power ratio between power and steering lines, from low to high power setting: red, green, blue, black. During the powering up there is a shift in the force distribution over the kite. The steering lin

---

## Author Response (AR1)

**Aerodynamic characterization of a soft kite by in situ flow measurement**

Johannes Oehler[1] and Roland Schmehl[1]

[revised manuscript text omitted]

in Table 1 allows to execute dynamic flight maneuvers  and handle kites with a wingspan of

$b = 10$m, produce 10 m or larger, at flight speeds above $20\,\mathrm{ms}^{-1}$ and withstand large pulling 20 m/s while withstanding tensile forces of several kilonewtons and more.

To avoid the uncertainty from an estimated wind speed for this testing method measuring the kN or more. It is the objective of the present study to develop an experimental method for aerodynamic characterization of large deformable membrane kites that are used for energy conversion. At the core of this method is a novel setup for the accurate measurement of the relative flow conditions at the flying kite is necessary. This is why in the experiment of this paper we installed a sensor setup for apparent flow magnitude and flow angles in the bridle system kite during energy-generation in pumping cycles. Since the setup is additional equipment for tests of a commercial prototype the mounting of the setup has to consume as little time as possible.

The paper is organized as follows. In Sect. 2 we describe the airborne components of the kite . Jann and Greiner-Perth (2017) developed a similar setup for a gliding parachute which measures power system, the measurement setup and the data acquisition procedure. In Sect. 3 we describe how the power setting is related to the angle of attack and flow velocity in the bridle lines between payload and canopy. By choosing such a setup that is independent from the ground station we have no limits in traction force and can measure power kites that produce much more lift force than usual sports kites of the wing and how the aerodynamic properties are derived from the measured data. In Sect. 4 the results are presented and discussed.

Position of the sensors on the AWES prototype: Tether force $F_t$ and reel-out speed $v_t$ are recorded at the ground station. GPS and IMU modules are mounted on the kite. The kite control unit steers the kite and measures current lengths of the steering and depower lines. Flow sensors for $\alpha_m$, $\beta_s$ and $v_a$ are mounted in the power lines that connect to the leading edge of the kite.

**2   System description and data acquisition**

The kite power research group of Delft University of Technology uses an AWES prototype operated by Kitepower B.V. The experimental study is based on the AWES prototype developed and operated by the company Kitepower as a test platform . The system within the EU Horizon 2020 "Fast Track to Innovation" project REACH (European Commission, 2015). The prototype can be classified as a ground generation pumping cycle AWES with a flexible wing kite . The power production is achieved in a cyclic flight pattern where the traction or power phase alternates with a retraction phase. During ground-generation AWES, operating a remote-controlled soft kite on a single tether. This general setup is illustrated schematically in Fig. 3 (right). The main system components are the ground station for converting the linear traction motion of the kite into electricity, the main tether and the C-shaped, bridled wing with the suspended kite control unit (KCU). In the following, we will denote the assembly of wing, bridle line system and KCU as "kite". To generate power the kite is operated in cyclic flight patterns with alternating traction and retraction phases. During the traction phase the kite flies dynamic crosswind maneuvers to produce a high pulling force and reel out the tether so the generator produces power. Retraction phase means that the performs crosswind maneuvers, such as figure-of-eight or circular flight patterns, while the tether is reeled off a drum that is connected to a generator. In this phase the AWES generates electricity. For the subsequent retraction phase the crosswind maneuvers are stopped and the generator is operated as a motor to reel in tether where low tether force is desired. During reel-in the kite is in a rather static flight

[revised manuscript text omitted]
_\mathrm{m}$-axis is aligned with the ~~two vertical bars of the airborne wind energy system on-board measurement equipment (AWESOME), y-axis is parallel to AWESOME's horizontal bar (see Fig. ??). X-axis points forward in flight direction and is normal to both the plane formed by the V-shaped front bridle lines and AWESOME's main structure. The power line bridles are assumed to be straight lines with negligible inertia as suggested in Bosch et al. (2013), the x-axis is thus always aligned with the kite's heading.~~

[Figure]

**Figure 5.** Front view (left) and side view (right) of the LEI V3 kite with reference frames, geometric parameters, mass distribution and definition of the reference chord $c_{\mathrm{ref}}$. The total wing surface area is denoted as $S$, while the projected value is denoted as $A$. The mass of the bridle lines is part of the wing mass. The side view distinguishes between the physical (real) kite and bridle line system, displayed in the background, and the overlaid simplified geometric depower model. The explicit dimensions describe the unloaded design shape of the wing.

 transverse member. Because the measurement frame is attached to the two tensioned power lines the $x_{\mathrm{m}}$-axis defines the heading of the kite. The rotation of the $x_{\mathrm{t}}$-axis into the $x_{\mathrm{m}}$-axis is described by the angle $\lambda_0$, which is not constant and can not be controlled actively. The angle depends on the aerodynamic load distribution acting on the wing, the kite design and the bridle layout. The inflow angles $\beta_{\mathrm{s}}$ and $\alpha_{\mathrm{m}}$ are determined in the measurement reference frame. Because the $z_{\mathrm{m}}$-axis can be regarded as the yaw axis of the kite, the inflow

angle $\beta_s$ is equivalent to the side slip angle. Similarly, the $y_m$-axis can be regarded as the pitch axis of the kite  and the inflow angle $\alpha_m$ is a measure for its pitch orientation with respect to the relative flow.

To transform $\alpha_m$ into a meaningful angle of attack  of the wing we define a reference chord
5  $c_{ref}$ which describes the pitch orientation of the wing within the kite system as a function of the symmetric actuation of the steering lines. This two-dimensional, simplified geometric depower model is illustrated in Fig. 5 (right). For the fully powered kite, the reference chord is defined to be perpendicular to the plane spanned by the power lines. Depowering the kite is modeled as a pitching of the reference chord around the front suspension point, while the real wing additionally deforms by spanwise twisting and bending. The specific bridle layout of the LEI V3 kite shifts
10 the front suspension point about 0.5 m backwards from the leading edge. The rotation is described by the depower angle $\alpha_d$ and by definition the fully powered state is given by $\alpha_d = 0$. A reference chord that is perpendicular to the power line plane is a reasonable approximation of the fully powered wing which is designed for optimal transfer of the aerodynamic load from the membrane wing to the bridle line system. These structural requirements are generally met best if the front bridle lines, which transmit most of the forces, connect perpendicularly to the wing. It is in principle straightforward to account for a constant
15 offset angle $\alpha_0$ (Fechner et al., 2015), however, for the investigated kite design this offset angle is rather small. For this reason we set $\alpha_0 = 0$.

The geometrical dimensions are extracted from the CAD geometry of the kite. The distance of the front suspension point from the bridle point is $d = 11.0$ m. For the fully powered kite, the distance of to the rear suspension point from the bridle point is $l_0 = 11.22$ m. The length of the reference chord can be determined as $c_{ref} = 2.2$ m. The kite is depowered by extending
20 the rear suspension of the wing by $\Delta l$. In the following section, we relate this length extension to the deployed length $l_d$ of the depower tape and the relative power setting $u_p$. The angle of attack of the relative flow with respect to the reference chord is calculated from the measured inflow angle and the depower angle as

25 $$\alpha = \alpha_m - \alpha_d, \tag{2}$$

while the angle of attack of the relative flow with respect to the tether reference frame is calculated as

$$\alpha_t = \alpha_m + \lambda_0. \tag{3}$$

30

Figure 6 illustrates how the azimuth angle $\phi$, the elevation angle $\beta$ and the radial distance $r$ are used to specify the position of the bridle point $\mathbf{B}$ relative to the ground attachment point $\mathbf{O}$.

[Figure]

**Figure 6.** Ground reference frame $(x_\mathrm{w}, y_\mathrm{w}, z_\mathrm{w})$, tether reference frame $(x_\mathrm{t}, y_\mathrm{t}, z_\mathrm{t})$, heading angle $\psi$ and spherical coordinates $(\beta, \phi, r)$. Only in case of a straight tether, the $z_\mathrm{t}$-axis is pointing in radial direction to the ground attachment point **O**.

 The heading angle $\psi$ specifies the orientation of  that produces lift distributed over its whole canopy we assume a relation between the power setting $u_p$ and the effective aerodynamic orientation of kite in the

**3.2**

$$\alpha = \alpha_m - \alpha_d.$$

 local tangential plane $\tau$. The angle is measured between the local upward direction (dotted line) and the projection of the $x_\mathrm{t}$-axis onto the tangential plane. Similarly, the course angle $\chi$ (not displayed) specifies the direction of the tangential kite velocity $\mathbf{v}_{\mathrm{k},\tau}$ in the local tangential plane. Combining Eqs. (2) and  we derive the relation

(3) to eliminate the measured inflow angle $\alpha_m$ we can differentiate three distinct contributions to the angle of attack

$$\cos(90^o + \alpha_d) = \frac{b^2 + c_{eff}^2 - (a + \Delta l)^2}{2dc_{eff}} \quad \alpha_t - \lambda_0 - \alpha_
[revised manuscript text omitted]